# MAGPIE: ALIGNMENT DATA SYNTHESIS FROM SCRATCH BY PROMPTING ALIGNED LLMS WITH NOTHING

**Zhangchen Xu**♠     **Fengqing Jiang** ♠     **Luyao Niu**♠     **Yuntian Deng** ◇
**Radha Poovendran**♠     **Yejin Choi**♠◇     **Bill Yuchen Lin**♠◇

♠University of Washington          ◇Allen Institute for AI

 https://magpie-align.github.io/    🤗 https://hf.co/magpie-align

## ABSTRACT

High-quality instruction data is critical for aligning large language models (LLMs). Although some models, such as Llama-3-Instruct, have open weights, their alignment data remain private, which hinders the democratization of AI. High human labor costs and a limited, predefined scope for prompting prevent existing open-source data creation methods from scaling effectively, potentially limiting the diversity and quality of public alignment datasets. Is it possible to synthesize high-quality instruction data at scale by extracting it directly from an aligned LLM? We present a *self-synthesis* method for generating large-scale alignment data named MAGPIE. Our key observation is that aligned LLMs like Llama-3-Instruct can generate a user query when we input only the pre-query templates up to the position reserved for user messages, thanks to their auto-regressive nature. We use this method to prompt Llama-3-Instruct and generate 4 million instructions along with their corresponding responses. We further introduce extensions of MAGPIE for filtering, generating multi-turn, preference optimization, domain-specific and multilingual datasets. We perform a comprehensive analysis of the MAGPIE-generated data. To compare MAGPIE-generated data with other public instruction datasets (e.g., ShareGPT, WildChat, Evol-Instruct, UltraChat, OpenHermes, Tulu-V2-Mix, GenQA), we fine-tune Llama-3-8B-Base with each dataset and evaluate the performance of the fine-tuned models. Our results indicate that using MAGPIE for supervised fine-tuning (SFT) solely can surpass the performance of previous public datasets utilized for both SFT and preference optimization, such as direct preference optimization with UltraFeedback. We also show that in some tasks, models supervised fine-tuned with MAGPIE perform comparably to the official Llama-3-8B-Instruct, despite the latter being enhanced with 10 million data points through SFT and subsequent preference optimization. This advantage is evident on alignment benchmarks such as AlpacaEval, ArenaHard, and WildBench.

## 1 INTRODUCTION

Large language models (LLMs) such as GPT-4 (Achiam et al., 2023) and Llama-3 (Meta, 2024) have become integral to AI applications due to their exceptional performance on a wide array of tasks by following instructions. The success of LLMs is heavily reliant on the data used for instruction fine-tuning, which equips them to handle a diverse range of tasks, including those not encountered during training. The effectiveness of instruction tuning depends crucially on access to high-quality instruction datasets. However, the alignment datasets used for fine-tuning models like Llama-3-Instruct are typically private, even when the model weights are open, which impedes the democratization of AI and limits scientific research for understanding and enhancing LLM alignment.

To address the challenges in constructing high-quality instruction datasets, researchers have developed two main approaches. The first type of method involves human effort to generate and curate instruction data (Databricks, 2023; Köpf et al., 2023; Zhao et al., 2024; Zheng et al., 2024; 2023), which is both *time-consuming* and *labor-intensive* (Liu et al., 2024a). In contrast, the second type of methods uses LLMs to produce synthetic instructions (Ding et al., 2023; Yin et al., 2023; Li et al., 2024a; Sun et al., 2023; Taori et al., 2023; Wang et al., 2023; 2024c; Xu et al., 2023a;b; Li et al., 2023a). Although these methods reduce human effort, its success heavily depends on prompt engineering

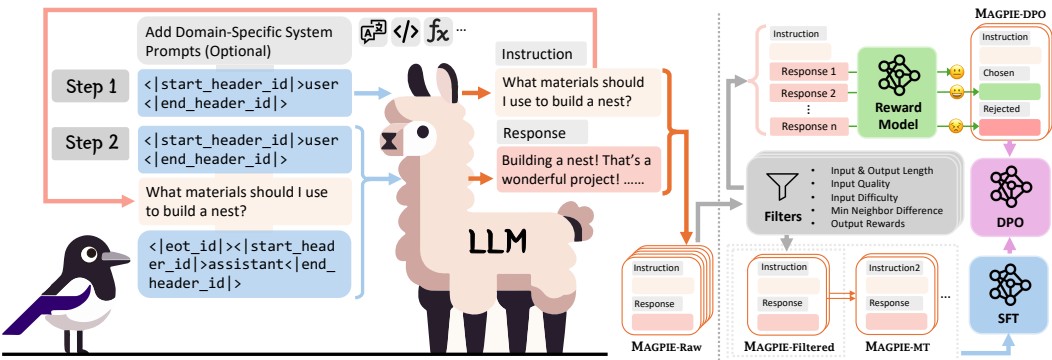

Figure 1: This figure illustrates MAGPIE, the process of self-synthesizing alignment data from aligned LLMs (e.g., Llama-3-8B-Instruct) to create a high-quality instruction dataset. In Step 1, we input only the pre-query template into the aligned LLM and generate an instruction along with its response using auto-regressive generation. In Step 2, we use a combination of a post-query template and another pre-query template to wrap the instruction generated from Step 1, prompting the LLM to generate the response. This completes the construction of the instruction dataset. MAGPIE efficiently generates diverse and high-quality instruction data, which can be further extended to multi-turn (MAGPIE-MT), preference optimization (MAGPIE-DPO), domain-specific, and multilingual datasets.

and the careful selection of initial seed questions. The *diversity* of synthetic data tends to decrease as the dataset size grows. Despite ongoing efforts, the scalable creation of high-quality and diverse instruction datasets continues to be a challenging problem.

*Is it possible to synthesize high-quality instructions at scale by directly extracting data from advanced aligned LLMs?* A typical input to an aligned LLM contains three key components: the pre-query template, the query, and the post-query template. For instance, an input to Llama-2-chat could be "`[INST]` Hi! `[/INST]`", where `[INST]` is the pre-query template and `[/INST]` is the post-query template. These templates are predefined by the creators of the aligned LLMs to ensure the correct prompting of the models. We observe that when we only input the pre-query template to aligned LLMs such as Llama-3-Instruct, they *self-synthesize* a user query due to their auto-regressive nature. Our experiments indicate that these random user queries are of high quality and great diversity, suggesting that the abilities learned during the alignment process are effectively utilized.

Based on these findings, we developed a self-synthesis method to construct high-quality instruction datasets at scale, named MAGPIE (as illustrated in Figure 1). Unlike existing methods, our approach does not rely on prompt engineering or seed questions. Instead, it *directly* constructs instruction data by prompting aligned LLMs with a pre-query template for sampling instructions. We also demonstrated the extensibility of MAGPIE in generating multi-turn, preference optimization, domain-specific, and multilingual datasets. We applied MAGPIE to the Llama-3-8B-Instruct and Llama-3-70B-Instruct models, creating two instruction datasets: MAGPIE-Air and MAGPIE-Pro, respectively.

Our MAGPIE-Air and MAGPIE-Pro datasets were created using 206 and 614 GPU hours, respectively, without any human intervention or API access to production LLMs like GPT-4. The statistics and advantages of MAGPIE datasets compared to existing ones are summarized in Table 4 in Appendix A. We perform a comprehensive analysis of these two datasets in Section 3, allowing practitioners to filter and select data instances for fine-tuning models according to their particular needs.

To compare MAGPIE data with other public instruction datasets (e.g., ShareGPT (Chiang et al., 2023), WildChat (Zhao et al., 2024), Evol Instruct (Xu et al., 2023a), UltraChat (Ding et al., 2023), OpenHermes (Teknium, 2023a;b), GenQA (Chen et al., 2024), Tulu V2 Mix (Ivison et al., 2023)), we conducted supervised fine-tuning (SFT) of the Llama-3-8B-Base model with each dataset and assess the performance of the fine-tuned models on alignment benchmarks such as AlpacaEval 2 (Li et al., 2023b), Arena-Hard (Li et al., 2024b), and WildBench (Lin et al., 2024). Our results show that models supervised fine-tuned with MAGPIE achieve superior performance, even surpassing models that utilize both SFT and direct preference optimization (DPO) (Rafailov et al., 2023) with UltraFeedback (Cui et al., 2023). Notably, MAGPIE-aligned models outperform the official Llama-3-8B-Instruct model on AlpacaEval 2, despite the latter being fine-tuned with over 10 million data points for SFT and subsequent preference optimization. Not only does MAGPIE excel in SFT alone compared to prior

public datasets, but also delivers the best results when combined with preference optimization methods such as DPO. By leveraging MAGPIE extensions to generate high-quality preference optimization datasets, MAGPIE-aligned Llama-3 models can even outperform GPT-4-Turbo(1106) on AlpacaEval 2. These findings show the exceptional quality of instruction data generated by MAGPIE, enabling it to outperform even the official, extensively optimized, and proprietary LLMs.

## 2 MAGPIE: A SCALABLE METHOD TO SYNTHESIZE ALIGNMENT DATA

**Chat Templates of Aligned LLMs.** For an aligned LLM (e.g., Llama-3-8B-Instruct), the input sequence can be represented as $x = T_{pre-query} \oplus q \oplus T_{post-query}$. Here, $q$ is the user query (e.g., "What material should I use to build a nest?"), while $T_{pre-query}$ and $T_{post-query}$ are pre-query and post-query templates. The pre-query template shows up before the user query, and the post-query template is defined as the conversation template between the user query and the LLM response. These templates are defined by the model provider to ensure the correct prompting. For example, for Llama-3-8B-Insturct model, $T_{pre-query}$ = `<|start_header_id|>user<|end_header_id|>`, and $T_{post-query}$ = `<|eot_id|><|start_header_id|>assistant<|end_header_id|>`.

### 2.1 MAGPIE PIPELINE

**Overview of MAGPIE.** In what follows, we describe our lightweight and effective method, MAGPIE, to synthesize alignment data from aligned LLMs. An instance of instruction data consists of at least one or multiple instruction-response pairs. Each pair specifies the roles of instruction provider (e.g., user) and follower (e.g., assistant), along with their instruction and response. As shown in Figure 1, MAGPIE consists of two steps: (1) instruction generation, and (2) response generation. The MAGPIE pipeline can be fully *automated without any human intervention*, and can be readily adapted for the generation of multi-turn, preference, and domain-specific datasets, as detailed in Section 2.2. We describe each step in the following.

**Step 1: Instruction Generation.** The goal of this step is to generate an instruction for each instance of instruction data. Given an open-weight aligned LLM (e.g., Llama-3-70B-Instruct), MAGPIE crafts a pre-query template in the format of the predefined instruction template of the LLM. Note that the auto-regressive LLM has been fine-tuned using instruction data in the format of the pre-query template. Thus, the LLM autonomously generates an instruction when the pre-query template crafted by MAGPIE is given as an input. MAGPIE stops generating the instruction once the LLM produces an end-of-sequence token. Sending the crafted query to the LLM multiple times leads to a set of instructions. We note that compared with existing synthetic approaches (Ding et al., 2023; Li et al., 2024a; Taori et al., 2023; Wang et al., 2023; 2024c; Xu et al., 2023a;b), MAGPIE does not require specific prompt engineering techniques since the crafted query follows the format of the predefined instruction template. In addition, MAGPIE autonomously generates instructions without using any seed question, ensuring the diversity of generated instructions.

**Step 2: Response Generation.** The goal of this step is to generate responses to the instructions obtained from Step 1. MAGPIE sends these instructions to the LLM to generate the corresponding responses. Combining the roles of instruction provider and follower, the instructions from Step 1, and the responses generated in Step 2 yields the instruction dataset. We note that separating instruction and response generation offers several key advantages, including *more flexible generation configurations* between instructions and responses, where instruction generation benefits from higher temperature settings to maximize diversity, while response generation requires lower temperature for accuracy and reliability. In addition, this approach provides *modular flexibility*, allowing users to generate instructions independently and later pair them with responses from various sources.

**Applicability of MAGPIE on Different LLMs.** MAGPIE can be readily deployed to state-of-the-art open-weight language models including but not limited to Llama-3 (Meta, 2024), Llama-3.1/3.3 (Dubey et al., 2024), Qwen2 (Yang et al., 2024a), Qwen2.5 (Yang et al., 2024b), Gemma-2 (Team et al., 2024), and Phi-3 (Abdin et al., 2024). Please refer to Appendix A for detailed support and corresponding datasets.

**Remark.** MAGPIE generates high-quality instructions even when the instruction loss is masked during alignment. We hypothesize that LLMs retain an implicit memorization of instruction distributions. We leave it as a potential future research problem.

## 2.2 MAGPIE EXTENSIONS

**Dataset Filtering.** MAGPIE allows practitioners to select instruction data from the raw dataset generated from the above two steps based on their needs. In Appendix C, we explores potential filter configurations with eight available metrics for users to customize their own MAGPIE datasets. We also provide 6 off-the-shelf filter configurations and discuss their performance in Appendix F.5.

**Generating Multi-Turn Instruction Datasets.** MAGPIE can be readily extended to generate multi-turn instruction datasets. To construct a multi-turn dataset (denoted as MAGPIE-MT), we initially follow Steps 1 and 2 to generate the first turn of instruction and response. For subsequent turns, we append the pre-query template to the end of the full prompt from the previous round of communication. We observe that the model may occasionally forget its role as the user, especially for the 8B model. To mitigate this, we employ a system prompt designed to control the behavior of the LLM and reinforce its awareness of the multi-round conversation context. The full prompt for building the instructions of MAGPIE-MT can be found in Figure 14 in Appendix G. We follow the procedure in Step 2 of Section 2.1 to generate responses to form the multi-turn instruction dataset.

**Generating Preference Optimization Datasets.** Leveraging the diverse and high-quality instructions produced by MAGPIE, we present a simple and effective method for generating preference optimization data, inspired by Meng et al. (2024) and Tran et al. (2023). We first select a small proportion of high-quality instructions from the raw dataset generated by the MAGPIE pipeline, ensuring diverse task categories. For each selected instruction, we sample responses from the aligned LLM $k$ times, using a temperature of $T < 1$. We then employ a reward model (RM) to annotate scores for these responses. The response with the highest RM score is labeled as the chosen response, while the one with the lowest RM score is designated as the rejected response.

**Generating Domain-Specific and Multilingual Datasets.** In certain scenarios, users may wish to fine-tune LLMs using domain-specific or multilingual instruction data to enhance performance within specific domains or languages. To address this need, we introduce a lightweight method to control both the task category and the language of generated instructions. Our approach involves guiding LLMs through a tailored system prompt, specifying that the model is a chatbot designed for a particular domain and outlining the types of user queries it might encounter. Examples of system prompts designed to control the generation of math, code, translation, and multilingual instructions are illustrated in Figure 15 in Appendix G.

Furthermore, we note that domain-specific and multilingual instruction data can also be generated using models that are tailored to particular fields. MAGPIE demonstrates broad applicability beyond diverse chat models, extending to specialized code models (e.g., DeepSeek-Coder-V2 (Zhu et al., 2024)) and math models (e.g., Qwen2-Math-7B-Instruct (Yang et al., 2024a)). By leveraging the unique strengths and specializations of different models, MAGPIE can create a rich and diverse corpus of instructions. Examples of MAGPIE-generated instructions from different domain-specific models and multilingual models are provided in Appendix I.

## 3 DATASET ANALYSIS

To demonstrate the effectiveness of MAGPIE compared with baseline methods for generating diverse high-quality alignment datasets, we apply MAGPIE to the Llama-3-8B-Instruct and Llama-3-70B-Instruct models (Meta, 2024) to construct two instruction datasets: Llama-3-MAGPIE-Air (hereafter referred to as MAGPIE-Air) and Llama-3-MAGPIE-Pro (hereafter referred to as MAGPIE-Pro), respectively. Examples of instances in both datasets can be found in Appendix I. In this section, we present a comprehensive analysis of the MAGPIE-Air and MAGPIE-Pro datasets, including topic coverage, difficulty, quality, similarity of instructions, and the quality of the responses.

### 3.1 DATASET COVERAGE

We follow Zhao et al. (2024) and analyze the coverage of MAGPIE-Pro in the embedding space. Specifically, we use the `all-mpnet-base-v2` embedding model[1] to calculate the input embeddings, and employ t-SNE (Van der Maaten & Hinton, 2008) to project these embeddings into a

---

[1] https://huggingface.co/sentence-transformers/all-mpnet-base-v2

two-dimensional space. We adopt three synthetic datasets as baselines, including **Alpaca** (Taori et al., 2023), **Evol Instruct** (Xu et al., 2023a), and **UltraChat** (Ding et al., 2023), to demonstrate the coverage of MAGPIE-Pro. The detailed analysis can be found in Appendix D.1. We observe that the t-SNE plot of MAGPIE-Pro encompasses the area covered by the plots of Alpaca, Evol Instruct, and UltraChat. This suggests that MAGPIE-Pro provides a broader or more diverse range of topics. We also follow Wang et al. (2023) and present the most common verbs and their top direct noun objects in instructions in Appendix D, indicating the diverse topic coverage of MAGPIE dataset.

## 3.2 DATASET ATTRIBUTES

**Attribute: Task Categories of Instructions.** We use Llama-3-8B-Instruct to categorize the instances in MAGPIE-Pro (see Figure 9 in Appendix D.1 for detail). The prompts used to query Llama-3-8B-Instruct can be found in Appendix G. Our observations indicate that over half of the tasks in MAGPIE-Pro pertain to information seeking, making it the predominant category. This is followed by tasks involving creative writing, advice seeking, planning, and math. This distribution over the task categories aligns with the practical requests from human users (Li et al., 2023b).

**Attribute: Quality of Instructions.** Similar to methods in (Chen et al., 2023), we prompt the Llama-3-8B-Instruct model to assess the quality of each instruction in MAGPIE-Air and MAGPIE-Pro, categorizing them as 'very poor', 'poor', 'average', 'good', and 'excellent'. We present the histograms of qualities for both datasets in Figure 2-(a). We have the following two observations. First, both datasets are of high quality, with the majority of instances rated 'average' or higher. In addition, the overall quality of MAGPIE-Pro surpasses that of MAGPIE-Air. We hypothesize that this is due to the enhanced capabilities of Llama-3-70B compared with Llama-3-8B.

**Attribute: Difficulty of Instructions.** We use the Llama-3-8B-Instruct model to rate the difficulty of each instruction in MAGPIE-Air and MAGPIE-Pro. Each instruction can be labeled as 'very easy', 'easy', 'medium', 'hard', or 'very hard'. Figure 2-(b) presents the histograms of the levels of difficulty for MAGPIE-Air and MAGPIE-Pro. We observe that the distributions across difficulty levels are similar for MAGPIE-Air and MAGPIE-Pro. Some instructions in MAGPIE-Pro are more challenging than those in MAGPIE-Air because MAGPIE-Pro is generated by a more capable model (i.e., Llama-3-70B-Instruct).

Figure 2: The statistics of instruction difficulty and quality.

**Attribute: Instruction Similarity.** We quantify the similarity among instructions generated by MAGPIE to remove repetitive instructions. We measure the similarity using **minimum neighbor distance** in the embedding space. Specifically, we first represent all instructions in the embedding space using the `all-mpnet-base-v2` embedding model. For any given instruction, we then calculate the minimum distance from the instruction to its nearest neighbors in the embedding space using Facebook AI Similarity Search (FAISS) (Douze et al., 2024). The minimum neighbor distances of instructions in MAGPIE-Air after removing repetitions are summarized in Figure 3-(a).

**Attribute: Quality of Responses.** We assess the quality of responses using **rewards** assigned by a reward model, denoted as $r^*$. For each instance in our dataset, we also calculate **reward difference** as $r^* - r_{base}$, where $r_{base}$ is the reward assigned by the same reward model to the

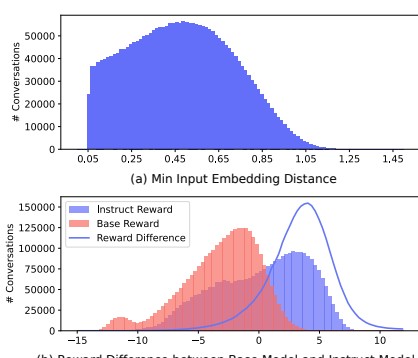

Figure 3: This figure summarizes the minimum neighbor distances and reward differences.

response generated by the Llama-3 base model for the same instruction. We use URIAL (Lin et al., 2023) to elicit responses from the base model. A positive reward difference indicates that the response from our dataset is of higher quality, and could potentially benefit instruction tuning. In our experiments, we follow Lambert et al. (2024) and choose `FsfairX-LLaMA3-RM-v0.1` (Xiong et al., 2024) as the reward model. Our results on the reward difference are presented in Figure 3-(b).

## 3.3 SAFETY ANALYSIS

We use Llama-Guard-2 (Team, 2024) to analyze the safety of MAGPIE-Air and MAGPIE-Pro. Our results indicate that both datasets are predominantly safe, with less than 1% of the data potentially containing harmful instructions or responses. Please see Appendix D.2 for detailed safety analysis.

## 3.4 COST ANALYSIS

We perform experiments on a server with four NVIDIA A100-SXM4-80GB GPUs, an AMD EPYC 7763 64-Core Processor, and 512 GB of RAM, using the VLLM inference framework (Kwon et al., 2023). The models are loaded in the `bfloat16` format.

When creating the 3M MAGPIE-Air dataset, our MAGPIE spent 1.55 and 50 hours to generate the instructions (Step 1) and responses (Step 2), respectively. For the 1M MAGPIE-Pro dataset, MAGPIE used 3.5 and 150 hours to generate the instructions (Step 1) and responses (Step 2), respectively. Compared to existing approaches to create instruction datasets, the pipeline of MAGPIE is fully automated without any human intervention or API access to advanced commercial models such as GPT-4 (Achiam et al., 2023). Consequently, MAGPIE is cost-effective and scalable. On average, implementing MAGPIE on a cloud server[2] would incur costs of **$0.12** and **$1.1** per 1,000 data instances for MAGPIE-Air and MAGPIE-Pro, respectively.

## 3.5 ADDITIONAL ANALYSIS

Additional dataset analysis, including the impact of generation configurations on the quality and difficulty of the generated instructions, is in Appendix D.3. Ablation analysis on annotating models for assessing quality and difficulty is in Appendix D.4. Contamination analysis is in Appendix D.5.

## 4 PERFORMANCE ANALYSIS

In this section, we evaluate the quality of MAGPIE-generated datasets by utilizing them to align base models including Llama-3 (Meta, 2024), Qwen1.5 (Bai et al., 2023), and Qwen2 (Yang et al., 2024a).

## 4.1 EXPERIMENTAL SETUPS

**Baselines for Supervised Fine-Tuning and Preference Optimization.** We compare the family of instruction datasets generated by MAGPIE with eight SOTA open-source instruction datasets: **ShareGPT** (Chiang et al., 2023), **WildChat** (Zhao et al., 2024), **Evol Instruct** (Xu et al., 2023a), **UltraChat** (Ding et al., 2023), **GenQA** (Chen et al., 2024), **OpenHermes 1** (Teknium, 2023a), **OpenHermes 2.5** (Teknium, 2023b), and **Tulu V2 Mix** (Ivison et al., 2023). ShareGPT and WildChat are representative human-written datasets containing 112K and 652K high-quality multi-round conversations between humans and GPT, respectively. Evol Instruct, UltraChat, and GenQA are representative open-source synthetic datasets. Following Meng et al. (2024), we use the 208K sanitized version of Ultrachat provided by HuggingFace[3]. OpenHermes 1, OpenHermes 2.5, and Tulu V2 Mix are crowd-sourced datasets consisting of a mix of diverse open-source instruction datasets, with 243K, 1M, and 326K conversations, respectively. We also create an instruction dataset with 100K conversations using the Self-Instruct (Wang et al., 2023) and Llama-3-8B-Instruct model, denoted as **Self-Instruct (Llama-3)**.

We compare the models aligned using MAGPIE with preference optimization baselines using direct preference optimization (DPO) (Rafailov et al., 2023). Specifically, we follow Meng et al. (2024)

---

[2] `https://lambdalabs.com/service/gpu-cloud`
[3] `https://huggingface.co/datasets/HuggingFaceH4/ultrachat_200k`

and use the models fine-tuned with the UltraChat dataset (for instruction tuning) and **Ultrafeedback** dataset (for preference optimization) (Cui et al., 2023).

**MAGPIE Setups.** To demonstrate the quality of MAGPIE-generated instruction datasets for SFT, we select the first 300K **MAGPIE-Air** and **MAGPIE-Pro** raw datasets generated by Llama-3-8B-Instruct and Llama-3-70B-Instruct models, respectively. Apart from these raw datasets, we also applied the filters detailed in Appendix C and created two filtered datasets: **MAGPIE-Air-Filtered** and **MAGPIE-Pro-Filtered**, each contains 300K conversations. For preference optimization, we generate two additional datasets: **MAGPIE-Air-DPO** (generated by Llama-3-8B-Instruct) and **MAGPIE-Pro-DPO** (generated by Llama-3-70B-Instruct) with $k = 5$ and $T = 0.8$, each contains 100K conversations. We use `RLHFlow/ArmoRM-Llama3-8B-v0.1` (Wang et al., 2024a) as the reward model.

**Model Alignment Details.** For supervised fine-tuning, we follow Touvron et al. (2023) and use a cosine learning rate schedule with an initial learning rate of $2 \times 10^{-5}$ when fine-tuning Llama-3, Qwen1.5 and Qwen2 base models. The maximum sequence length is 8192. For DPO, we use a cosine learning rate of $5 \times 10^{-7}$. The detailed parameters can be found in Appendix E.2. We follow the official instruction templates of each model.

**Evaluation Benchmarks.** We evaluate the performance of the aligned models using two widely adopted instruction-following benchmarks: AlpacaEval 2 (Li et al., 2023b) and Arena-Hard (Li et al., 2024b). AlpacaEval 2 consists of 805 representative instructions chosen from real user interactions. Arena-Hard is an enhanced version of MT-Bench (Zheng et al., 2023), containing 500 challenging user queries. Both benchmarks employ a GPT evaluator to assess responses generated by the model of interest and a baseline model. Specifically, we use GPT-4-Turbo (1106) and Llama-3-8B-Instruct as baselines for AlpacaEval 2. By default, Arena-Hard uses GPT-4 (0314) as its baseline model.

**Metrics.** We adopt two metrics to measure the capabilities of instruction-following of fine-tuned models. The first metric is the **win rate (WR)**, which calculates the fraction of responses that are favored by the GPT evaluator. This metric is applied in both benchmarks including AlpacaEval 2 and Arena-Hard. The second metric is the **length-controlled win rate (LC)** (Dubois et al., 2024), a debiased version of WR. The GPT evaluator considers the lengths of responses generated by the baseline model and model under evaluation when computing LC. By accounting for response length, LC reduces its impact on the win rate. This metric is specifically applied to the AlpacaEval 2 benchmark (Li et al., 2023b).

**More Experimental Setups.** We provide more detailed descriptions of our experimental setups, including more model alignment details and benchmark decoding hyper-parameters in Appendix E.

## 4.2 EXPERIMENTAL RESULTS

**MAGPIE datasets outperform baselines with SFT only.** In Table 1, we compare the performance of Llama-3 models fine-tuned with instruction datasets generated by MAGPIE against those supervised fine-tuned with baseline datasets. Using the AlpacaEval 2 benchmark, we observe that both the LC and WR of our supervised fine-tuned models surpass all those models fine-tuned with baseline SFT datasets. This indicates that the datasets generated by MAGPIE are of higher quality, leading to significantly enhanced instruction-following capabilities. A similar observation is made when using the Arena-Hard evaluation benchmark. We highlight that the Llama-3 base models supervised fine-tuned with instruction datasets generated by MAGPIE outperform even those models that have undergone preference optimization (i.e., STF followed by DPO), which further emphasizes the high quality of data generated by MAGPIE.

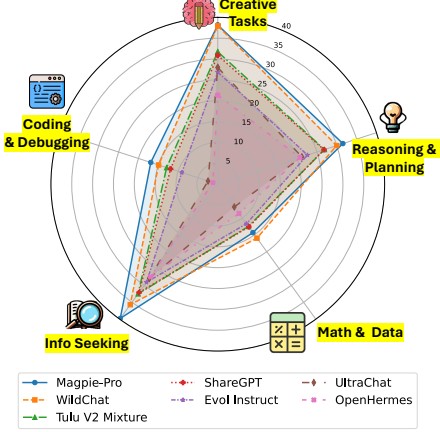

Figure 4: This figure shows the performance breakdown by category of MAGPIE-Pro and baselines on WildBench.

---

[4]`https://huggingface.co/meta-llama/Meta-Llama-3-8B-Instruct`

Table 1: This table compares the performance of models instruction-tuned on the Llama-3-8B base models using MAGPIE-generated datasets and baseline datasets. We observe that models aligned with our datasets significantly outperform those aligned with baseline datasets of the same order of magnitude in terms of data size. In addition, our fine-tuned models achieve comparable performance to the official aligned model, despite only undergoing SFT with a much smaller dataset. Numbers in **bold** indicate that MAGPIE outperforms the official Llama-3-8B-Instruct model.

| | Alignment Setup (Base LLM = **Llama-3-8B**) | #Convs | AlpacaEval 2 | | | | | | Arena-Hard |
|---|---|---|---|---|---|---|---|---|---|
| | | | GPT-4-Turbo (1106) | | | Llama-3-8B-Instruct | | | |
| | | | LC (%) | WR (%) | SD | LC (%) | WR (%) | SD | WR(%) |
| SFT | +Self-Instruct (Llama-3) (Wang et al., 2023) | 100K | 7.21 | 5.18 | 0.7 | 17.86 | 12.73 | 1.05 | 4.0 |
| | +ShareGPT (Chiang et al., 2023) | 112K | 9.73 | 7.2 | 0.81 | 27.26 | 18.32 | 1.18 | 6.5 |
| | +Evol Instruct (Xu et al., 2023a) | 143K | 8.52 | 6.25 | 0.76 | 20.16 | 14.98 | 1.1 | 5.1 |
| | +OpenHermes 1 (Teknium, 2023a) | 243K | 9.94 | 6.27 | 0.73 | 29.19 | 17.92 | 1.16 | 4.4 |
| | +Tulu V2 Mix (Ivison et al., 2023) | 326K | 9.91 | 7.94 | 0.86 | 24.28 | 18.64 | 1.18 | 5.4 |
| | +WildChat (Zhao et al., 2024) | 652K | 14.62 | 10.58 | 0.92 | 34.85 | 26.57 | 1.32 | 8.7 |
| | +OpenHermes 2.5 (Teknium, 2023b) | 1M | 12.89 | 9.74 | 0.91 | 32.68 | 25.01 | 1.30 | 8.2 |
| | +GenQA (Chen et al., 2024) | 6.47M | 9.05 | 7.11 | 0.82 | 21.90 | 16.09 | 1.12 | 3.0 |
| | +UltraChat (Ding et al., 2023) ↘ | 208K | 8.29 | 5.44 | 0.71 | 23.95 | 15.12 | 1.11 | 3.6 |
| + DPO | +UltraFeedback((Cui et al., 2023)) | 64K | 18.36 | 17.33 | 1.14 | 44.42 | 42.36 | 1.46 | 14.8 |
| SFT | **MAGPIE**-Air-300K-Raw | 300K | 21.99 | 21.65 | 1.21 | 48.63 | 48.06 | 1.42 | 15.8 |
| | **MAGPIE**-Air-300K-Filtered ↘ | 300K | 22.66 | **23.99** | 1.24 | 49.27 | **50.8** | 1.44 | 14.9 |
| + DPO | +**MAGPIE**-Air-DPO | 100K | **45.48** | **50.43** | 1.48 | **75.06** | **79.64** | 1.18 | **35.9** |
| SFT | **MAGPIE**-Pro-300K-Raw | 300K | 21.65 | 22.19 | 1.2 | 49.65 | **50.84** | 1.42 | 15.9 |
| | **MAGPIE**-Pro-300K-Filtered ↘ | 300K | **25.08** | **29.47** | 1.35 | **52.12** | **53.43** | 1.44 | 18.9 |
| + DPO | +**MAGPIE**-Pro-DPO | 100K | **50.10** | **53.53** | 1.45 | **78.52** | **80.82** | 1.17 | **35.7** |
| | Llama-3-8B-Instruct (SFT+DPO) | >10M[4] | 22.92 | 22.57 | 1.26 | 50 | 50 | - | 20.6 |

To investigate the advantages of MAGPIE across different task categories, we compare the performance of models fine-tuned with MAGPIE-Pro with baseline datasets using WildBench benchmark (Lin et al., 2024). This benchmark consists of 1024 tasks carefully selected from real-world human-LLM conversation logs. The results are demonstrated in Figure 4. We observe that MAGPIE consistently outperforms baseline datasets across categories.

**Models aligned with data generated by MAGPIE achieve comparable or even higher performance to the official aligned model, but with fewer data.** In Table 1, we also compare the performance of models aligned with data generated by MAGPIE against the official aligned model (Llama-3-8B-Instruct). We observe that the Llama-3-8B base model supervised fine-tuned with data from MAGPIE outperforms Llama-3-8B-instruct using the AlpacaEval 2 benchmark. For example, when Llama-3-8B-Instruct is chosen as the baseline model of AlpacaEval 2, we observe that LC of Llama-3-8B base models fine-tuned with instruction data from MAGPIE exceeds 50%, indicating a preference for our SFT models over the official aligned model. In addition, when DPO is applied, our aligned model demonstrates remarkable performance gains. Specifically, it outperforms the official Llama-3-8B-Instruct model on both the AlpacaEval 2 and Arena-Hard benchmarks. Most notably, our model even surpasses GPT-4-Turbo(1106) on AlpacaEval 2. Finally, we highlight that our alignment process uses no more than 400K data, whereas the official aligned models are aligned with more than 10M data samples. This demonstrates the high quality of the data generated by MAGPIE.

**MAGPIE can enhance the performance of other backbone models.** Table 2 illustrates the efficacy of MAGPIE when applied to generate instruction dataset and fine-tune other base models, i.e., Qwen2-1.5B, Qwen1.5-4B, and Qwen1.5-7B. The results demonstrate that our fine-tuned models achieve better performance than the official aligned models, which have undergone both supervised fine-tuning and preference tuning. These results underscore the effectiveness of MAGPIE and the quality of its generated instructions. In addition, we apply MAGPIE-generated datasets to align Llama-3.1-Minitron-4B-Width-Base (Sreenivas et al., 2024) and Llama-3.1-8B-Instruct (Dubey et al., 2024) using SFT followed by DPO. The resulting aligned model, which we term MagpieLM, achieves remarkable performance and ranks first among popular open-source instruction models with fewer than 10 billion parameters. The details of MagpieLM are deferred to Appendix B.

**Performance of MAGPIE on More Benchmarks.** We report the performance of models supervised fine-tuned using MAGPIE-Air and MAGPIE-Pro, evaluated across a range of tasks featured on the Huggingface Open LLM Leaderboard (Beeching et al., 2023) in Table 3. The tasks includes MMLU (Hendrycks et al., 2020), ARC Challenge (Clark et al., 2018), HellaSwag (Zellers et al., 2019),

Table 2: This table compares the performance of models instruction-tuned on the Qwen base models using the MAGPIE-Pro-300K-Filtered dataset and the official instruction-tuned models. The Qwen base model enhanced with MAGPIE outperforms the official instruction-tuned model.

| | | AlpacaEval 2 | | | | | |
| | | GPT-4-Turbo (1106) | | | Official Aligned Model as Ref. | | |
| Alignment Setup | | LC (%) | WR (%) | SD | LC (%) | WR (%) | SD |
|---|---|---|---|---|---|---|---|
| Qwen2-1.5B | Qwen2-1.5B-Instruct | **3.91** | 3.00 | 0.54 | 50 | 50 | - |
| | Base Model + MAGPIE | 3.48 | **5.32** | 0.67 | **56.66** | **66.27** | 1.50 |
| Qwen1.5-4B | Qwen1.5-4B-Chat | 5.89 | 4.74 | 0.67 | 50 | 50 | - |
| | Base Model + MAGPIE | **9.1** | **10.96** | 0.93 | **68.09** | **72.42** | 1.42 |
| Qwen1.5-7B | Qwen1.5-7B-Chat | 14.75 | 11.77 | 0.97 | 50 | 50 | - |
| | Base Model + MAGPIE | **15.10** | **18.51** | 1.14 | 46.28 | **58.53** | 1.44 |

Table 3: This table compares the performance of models supervised-fine-tuned on MAGPIE-Air, MAGPIE-Pro, and MAGPIE-Pro-Mix against baselines and official instruct model across various downstream benchmarks. All models are supervised-fine-tuned on the Llama-8B base models.

| Alignment Setup | MMLU (5) | ARC (25) | HellaSwag (10) | TruthfulQA (0) | WinoGrande (5) | GSM8K (5) | MMLU-Redux (0) | Average |
|---|---|---|---|---|---|---|---|---|
| ShareGPT | 66.03 | 58.45 | 81.50 | 52.34 | 74.03 | 48.67 | 50.68 | 61.67 |
| Evol Instruct | 65.62 | 60.75 | 82.70 | 52.87 | 76.16 | 42.91 | 52.73 | 61.96 |
| GenQA | 63.45 | 58.53 | 79.65 | 48.85 | 74.03 | 43.14 | 51.87 | 59.93 |
| OpenHermes 1 | 65.42 | 62.29 | 82.15 | 50.85 | 75.61 | 47.16 | 46.07 | 61.36 |
| OpenHermes 2.5 | 65.70 | 61.86 | 82.53 | 51.35 | 76.09 | 67.02 | 46.07 | **66.24** |
| Tulu V2 Mix | 66.34 | 59.22 | 82.80 | 47.99 | 76.16 | 58.07 | 46.97 | 62.51 |
| WildChat | 65.95 | 59.22 | 81.39 | 53.18 | 75.30 | 48.75 | 52.59 | 62.34 |
| UltraChat | 65.23 | 62.12 | 81.68 | 52.76 | 75.53 | 50.57 | 50.75 | 62.66 |
| MAGPIE-Air-300K-Filtered | 64.45 | 61.01 | 79.90 | 53.48 | 72.38 | 52.24 | 52.34 | 62.25 |
| MAGPIE-Pro-300K-Filtered | 64.25 | 60.41 | 80.52 | 52.46 | 73.32 | 47.92 | 52.16 | 61.58 |
| **MAGPIE-Pro-Mix-Filtered** | 65.65 | 59.64 | 80.72 | 50.81 | 73.24 | 63.08 | 56.34 | **64.21** |
| Llama-3-8B-Instruct | 67.82 | 61.52 | 78.67 | 52.47 | 72.14 | 71.72 | 58.60 | **66.13** |

TruthfulQA (Lin et al., 2021), WinoGrande (Levesque et al., 2012), and GSM8K (Cobbe et al., 2021). We also perform experiments on MMLU-Redux (Gema et al., 2024) with zero-shot prompting. Our experimental results demonstrate that models fine-tuned with MAGPIE-Air and MAGPIE-Pro achieve comparable performance to the official instruct model and other baselines.

We note that the performance of MAGPIE may degrade on reasoning tasks, which is attributed to the small proportion of reasoning instructions in MAGPIE-Air and MAGPIE-Pro datasets. In response, we provide a supplementary "booster" dataset containing 150K math, code, and reasoning instructions using the MAGPIE extension mentioned in Section 2.2. We combine this booster dataset with MAGPIE-Pro-300K-Filtered and create MAGPIE-Pro-Mix-Filtered. Experimental results presented in Table 3 demonstrate that the model supervised fine-tuned using the mixed dataset effectively addresses the initial weakness in reasoning tasks. Notably, this new model ranks among the top-3 of all model checkpoints, performing only slightly weaker than OpenHermes 2.5 (1M conversations) and Llama-3-8B-Instruct (>10M conversations). This significant improvement showcases the flexibility and adaptability of the MAGPIE framework in generating task-specific instruction data.

**Additional Experimental Results.** We defer additional experimental results and analysis of multi-turn datasets, i.e., MAGPIE-Air-MT and MAGPIE-Pro-MT, to Appendix F.1. We conduct a detailed comparison between MAGPIE and Self-Instruct in Appendix F.2. We demonstrate the performance of domain-specific and multi-lingual MAGPIE datasets in Appendix F.3. In addition, ablations on data quantity, quality, filter designs, and response generator are deferred in Appendices F.4, F.5, and F.6. MAGPIE model's performance on trustworthiness and instruction-following benchmarks is reported in Appendices F.7 and F.8. The ablation analysis on the impact of reward models on DPO data performance is presented in Appendix F.9.

## 5   RELATED WORK

**LLM Alignment.** Instruction tuning (Wei et al., 2022) and preference tuning (Bai et al., 2022) are widely used to align the responses of LLMs with human values. Instruction tuning utilizes an instruction dataset to fine-tune LLMs, where each instruction data consists of one turn or multiple turns of instructions and desired responses. The performance of instruction tuning heavily relies on

the quality of instruction data (Taori et al., 2023; Wang et al., 2023; Zhou et al., 2023a). Preference tuning further improves responses of LLMs using reinforcement learning human feedback (RLHF) (Bai et al., 2022) or preference optimization (Azar et al., 2024; Ethayarajh et al., 2024; Hong et al., 2024; Rafailov et al., 2023) based on a preference dataset.

**Alignment Dataset Construction.** We classify the existing methods of creating datasets for model alignment into two main categories: human interactions with LLMs and synthetic instruction generation. To create datasets for alignment, previous studies have collected **human** interactions with LLMs (Databricks, 2023; Zhao et al., 2024; Zheng et al., 2024; 2023; Köpf et al., 2023). However, manually crafting instructions is not only time-consuming and labor-intensive, but may also incorporate toxic content (Zhao et al., 2024). Another category of approaches (Wang et al., 2023; Taori et al., 2023; Xu et al., 2023a;b; Wang et al., 2024c; Sun et al., 2023; Wu et al., 2024) focus on prompting LLMs to generate **synthetic** instruction datasets, beginning with a small set of human-annotated seed instructions and expanding these through few-shot prompting. However, these methods face a diversity challenge, as few-shot prompting often results in new instructions that are too similar to the original seed questions (Li et al., 2024a). To enhance coverage, some research (Ding et al., 2023; Li et al., 2024a) summarizes world knowledge and employs it to generate synthetic datasets. We note that our MAGPIE dataset also belongs to the synthetic dataset. However, we leverage the prompt template without any requirement for seed questions or prompt engineering.

Compared to the above two main categories, alignment data can also be generated by **transforming** existing data (Wang et al., 2022; Sanh et al., 2022; Gandhi et al., 2024). However, the constrained variety of NLP tasks in these datasets may impede the ability of tuned LLMs to generalize in real-world scenarios (Li et al., 2024a). There are also **mixture** datasets (e.g., (Ivison et al., 2023; Teknium, 2023a; Liu et al., 2024b; Zhou et al., 2023a)) that combine or select high-quality instruction data from various existing open-source instruction datasets to enhance coverage (Ivison et al., 2023; Teknium, 2023a) and/or improve overall performance (Liu et al., 2024b; Zhou et al., 2023a). There are also data synthesis methods focusing on improving reasoning and math abilities (Yue et al., 2023; 2024), which can be further merged with MAGPIE for creating a better mixture of data for instruction tuning.

**Training Data Extraction.** Language models have the capability to memorize examples from their training datasets, potentially enabling malicious users to extract private information (Brown et al., 2022; Biderman et al., 2023; Carlini et al., 2021). Pioneering work (Krishna et al., 2020; Carlini et al., 2021; Nasr et al., 2023) has demonstrated that it is possible to extract private pre-training data from BERT (Devlin et al., 2018), GPT-2 (Radford et al., 2018), and ChatGPT (Achiam et al., 2023), respectively. Yu et al. (2023) propose several techniques including adjusting sampling strategies to better extract training datasets from language models. Recently, Kassem et al. (2024) propose a black-box prompt optimization method that uses an attacker LLM to extract high levels of memorization in a victim LLM. Wang et al. (2024b) leverage membership inference attack (MIA) to extract fine-tuning datasets from fine-tuned language models. Bai et al. (2024) extract the training dataset of production language models via special characters (e.g., structural symbols of JSON files, and , # in emails and online posts). Different from the prior work, we aim to create publicly available alignment datasets with minimal human effort by leveraging the remarkable generation capabilities of LLMs, rather than extracting private training data from LLMs.

## 6 CONCLUSION

In this paper, we developed a scalable method, MAGPIE, to synthesize instruction data for fine-tuning large language models. MAGPIE leveraged the predefined instruction templates of open-weight LLMs and crafted a prompt specifying only the role of instruction provider. Given the crafted prompt, the LLM then generated detailed instructions due to their auto-regressive nature. MAGPIE then sent the generated instructions to the LLM to generate corresponding responses. These pairs of instructions and responses constituted the instruction dataset. We used Llama-3-8B-instruct to label the instruction dataset and developed a filtering technique to select effective data instances for instruction tuning. We fine-tuned the Llama-3-8B base model using the selected data, and demonstrated that the fine-tuned model outperformed those fine-tuned using all baselines. Moreover, our fine-tuned models outperformed the official aligned model, Llama-3-8B-Instruct, which has been instruction-tuned and preference-optimized using more than 10M data instances. This highlighted the quality of the instruction data synthesized by MAGPIE.

## LIMITATIONS, DISCUSSIONS, AND ETHICAL CONSIDERATIONS

**Limitations and Discussions.** MAGPIE-aligned LLMs demonstrate strong performance on instruction following benchmarks compared to the official Llama-3-8B-Instruct. However, we observe a performance degradation on math and reasoning benchmarks. Although we leverage the techniques described in Section 2.2 to generate specialized booster reasoning datasets, there is still a performance gap between MAGPIE-aligned LLMs and the official models. Enhancing the reasoning ability of MAGPIE-aligned models presents a promising direction for future research.

**Societal Impact and Potential Harmful Consequences.** The primary objective of this paper is to develop a scalable method to synthesize instruction data to enhance the instruction-following capabilities of LLMs, and thus align them with human values. However, the data generated by MAGPIE may contain harmful instructions and/or responses, which may lead to unsafe behaviors if used raw in instruction tuning. Our empirical evaluations indicate that such harmful data instances constitute less than 1% of the dataset. Our data filtering technique in Appendix C can identify and remove these instances, thus mitigating the risk.

## ACKNOWLEDGMENT

The research of Z. Xu, F. Jiang, L. Niu, and R. Poovendran is partially supported by the Air Force Office of Scientific Research (AFOSR) under grant FA9550-23-1-0208, the National Science Foundation (NSF) AI Institute for Agent-based Cyber Threat Intelligence and Operation (ACTION) under grant IIS 2229876, and the Office of Naval Research (ONR) under grant N0014-23-1-2386. The research of Y. Choi is partially supported by NSF under grant DMS-2134012, IARPA HIATUS under grant 2022-22072200003, and ONR under grant N00014-24-1-2207. Results presented in this paper were partially obtained using the Chameleon testbed Keahey et al. (2020) supported by the National Science Foundation.

This work is supported in part by funds provided by the National Science Foundation, Department of Homeland Security, and IBM. Any opinions, findings, and conclusions or recommendations expressed in this material are those of the author(s) and do not necessarily reflect the views of the NSF or its federal agency and industry partners.

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

## A    STATISTICS OF INSTRUCTION DATASETS GENERATED BY MAGPIE COMPARED TO OTHER INSTRUCTION DATASETS.

MAGPIE can be readily deployed to state-of-the-art open-weight model families, including but not limited to Llama-3 (Meta, 2024), Llama-3.1/3.3 (Dubey et al., 2024), Qwen2 (Yang et al., 2024a), Qwen2.5 (Yang et al., 2024b), Gemma-2 (Team et al., 2024), and Phi-3 (Abdin et al., 2024) models.

In what follows, we compare datasets generated by MAGPIE with the above model families compared to other state-of-the-art instruction datasets. The MAGPIE dataset family encompasses over 13.4 million diverse and high-quality instructions and corresponding responses generated from state-of-the-art open-source models. This corpus represents the largest alignment dataset for LLMs that does not rely on human-written questions or employ complex multi-stage pipelines.

Table 4: Statistics of MAGPIE family compared to other instruction datasets. Tokens are counted using the `tiktoken` library (OpenAI, 2024). Links to the MAGPIE datasets are provided in the text.

| Instruction Source | Dataset Name | #Convs | #Turns | Human Effort | Response Generator | #Tokens / Turn | #Total Tokens |
|---|---|---|---|---|---|---|---|
| Synthetic | Alpaca (Taori et al., 2023) | 52K | 1 | Low | text-davinci-003 | $67.38_{\pm54.88}$ | 3.5M |
| | Evol Instruct (Xu et al., 2023a) | 143K | 1 | Low | ChatGPT | $473.33_{\pm330.13}$ | 68M |
| | UltraChat (Ding et al., 2023) | 208K | 3.16 | Low | GhatGPT | $376.58_{\pm177.81}$ | 238M |
| Human | Dolly (Databricks, 2023) | 15K | 1 | High | ChatGPT | $94.61_{\pm135.84}$ | 1.42M |
| | ShareGPT (Zheng et al., 2023) | 112K | 4.79 | High | ChatGPT | $465.38_{\pm368.37}$ | 201M |
| | WildChat (Zhao et al., 2024) | 652K | 2.52 | High | GPT-3.5 & GPT-4 | $727.09_{\pm818.84}$ | 852M |
| | LMSYS-Chat-1M (Zheng et al., 2024) | 1M | 2.01 | High | Mix | $260.37_{\pm346.97}$ | 496M |
| Mixture | Deita (Liu et al., 2024b) | 9.5K | 22.02 | - | Mix | $372.78_{\pm182.97}$ | 74M |
| | OpenHermes (Teknium, 2023a) | 243K | 1 | - | Mix | $297.86_{\pm258.45}$ | 72M |
| | Tulu V2 Mixture (Ivison et al., 2023) | 326K | 2.31 | - | Mix | $411.94_{\pm447.48}$ | 285M |
| **MAGPIE** | MAGPIE-Llama-3-Air | **3M** | 1 | **No** | Llama-3-8B-Instruct | $426.39_{\pm217.39}$ | 1.28B |
| | MAGPIE-Llama-3-Air-MT | 300K | 2 | **No** | Llama-3-8B-Instruct | $610.80_{\pm90.61}$ | 366M |
| | MAGPIE-Llama-3-Pro | 1M | 1 | **No** | Llama-3-70B-Instruct | $478.00_{\pm211.09}$ | 477M |
| | MAGPIE-Llama-3-Pro-MT | 300K | 2 | **No** | Llama-3-70B-Instruct | $554.53_{\pm133.64}$ | 333M |
| | MAGPIE-Llama-3.1-Pro | 1M | 1 | **No** | Llama-3.1-70B-Instruct | $482.35_{\pm378.45}$ | 482M |
| | MAGPIE-Llama-3.1-Pro-MT | 300K | 2 | **No** | Llama-3.1-70B-Instruct | $552.53_{\pm325.49}$ | 331M |
| | MAGPIE-Llama-3.3-Pro | 1M | 1 | **No** | Llama-3.3-70B-Instruct | $568.59_{\pm391.54}$ | 569M |
| | MAGPIE-Qwen2-Air | 3M | 1 | **No** | Qwen2-7B-Instruct | $577.87_{\pm416.10}$ | 1.73B |
| | MAGPIE-Qwen2-Pro | 1M | 1 | **No** | Qwen2-72B-Instruct | $424.87_{\pm339.71}$ | 424M |
| | MAGPIE-Qwen2.5-Pro | 1M | 1 | **No** | Qwen2.5-72B-Instruct | $693.31_{\pm271.45}$ | 693M |
| | MAGPIE-Gemma2-Pro | 534K | 1 | **No** | Gemma-2-27b-it | $483.90_{\pm237.80}$ | 259M |
| | MAGPIE-Phi3-Pro | 1M | 1 | **No** | Phi-3-Medium-Instruct | $391.38_{\pm414.32}$ | 391M |

## B    MAGPIELM

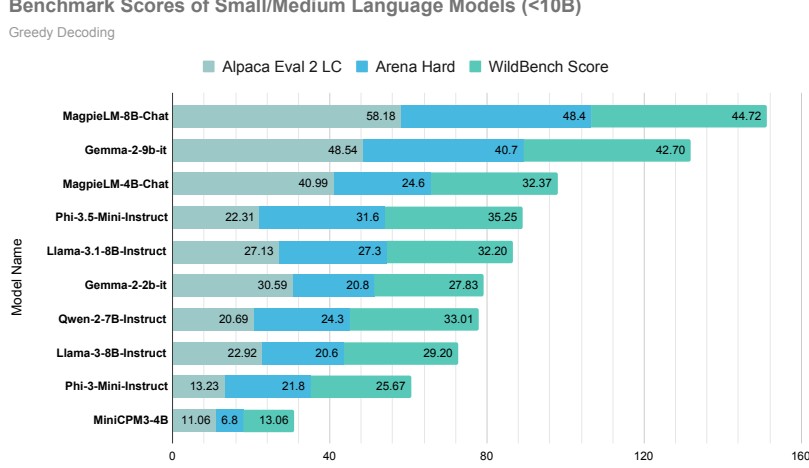

Figure 5: This figure shows the performance of MAGPIELM-4B-Chat and MAGPIELM-8B-Chat compared with baselines. MAGPIELM significantly outperforms baselines of similar sizes.

In this section, we discuss the details of our MAGPIELM. To construct the SFT and DPO datasets for aligning MAGPIELM, we select 550K and 200K high-quality instructions, respectively, from the MAGPIE family. These instructions cover diverse categories, ensuring a comprehensive training set. We then generate corresponding responses using the Gemma-2-9b-it model (Team et al., 2024).

The benchmark performance of both models is demonstrated in Figure 5. Notably, MAGPIELM significantly outperforms other baselines of similar model sizes across multiple benchmarks, including Alpaca Eval 2 (Li et al., 2023b), Arena Hard (Li et al., 2024b), and Wildbench (Lin et al., 2024).

## C  FILTER SETUPS

In this section, we explore potential filter configurations for selecting high-quality instructional data for fine-tuning purposes. We provide the following metrics to enable users to customize their filtered MAGPIE dataset:

1. **Input Length**: The total number of characters in the instructions.
2. **Output Length**: The total number of characters in the responses.
3. **Task Category**: The specific category of the instructions. See Appendix D.1 for details.
4. **Input Quality**: The clarity, specificity, and coherence of the instructions, rated as 'very poor', 'poor', 'average', 'good', and 'excellent'.
5. **Input Difficulty**: The level of knowledge required to address the task described in the instruction, rated as 'very easy', 'easy', 'medium', 'hard', or 'very hard'.
6. **Minimum Neighbor Distance**: The embedding distance to the nearest neighbor. Can be used for filtering out repetitive or similar instances.
7. **Reward**: Denoted as $r^*$. See Section 3 for details. This metric can be used to filter out low-quality responses, such as repetitions or refusals.
8. **Reward Difference**: Denoted as $r^* - r_{base}$. See Section 3 for details.

We provide several off-the-shelf configurations, as demonstrated in Table 5. We defer the detailed performance analysis of each filter configuration for MAGPIE-Pro to Appendix F.5.

Table 5: Different filter configurations we provide. We note that the Output Length filter is applied last. Specifically, this filter selects the $k$ instances of the longest responses. In our experiments, we empirically set $\tau_1 = -12$, and $\tau_2 = 0$.

| Source Dataset | Filter Name | #Convs | Input Length | Output Length | Task Category | Input Quality | Input Difficulty | Min Neighbor Distance | Reward | Reward Difference |
|---|---|---|---|---|---|---|---|---|---|---|
| MAGPIE-Air | Filter | 300K | - | Longest | - | ≥ good | ≥ medium | > 0 | - | > $\tau_2$ |
| MAGPIE-Pro | Filter | 300K | - | Longest | - | ≥ average | - | > 0 | > $\tau_1$ | - |
| | Filter2 | 300K | - | Longest | - | ≥ good | ≥ easy | > 0 | > $\tau_1$ | - |
| | Filter3 | 300K | - | Longest | - | - | - | > 0 | > $\tau_1$ | - |
| | Filter4 | 300K | - | Longest | - | ≥ good | ≥ easy | > 0 | - | > $\tau_2$ |
| | Filter5 | 338K | - | - | - | ≥ good | ≥ easy | > 0 | > $\tau_1$ | - |
| | Filter6 | 200K | - | Longest | - | - | 50% easy + 50% > easy | > 0 | > $\tau_1$ | - |

## D  MORE DATASET ANALYSIS

This section provides additional dataset analysis, complementing the discussions in Section 3. Statistics including lengths of instructions and responses are illustrated in Figure 6.

### D.1  ADDITIONAL ANALYSIS ON DATASET COVERAGE AND ATTRIBUTES.

**Dataset Coverage Measured by T-SNE and UMAP.**  Figure 7 presents the t-SNE and UMAP plots of MAGPIE, Alpaca, Evol Instruct, and UltraChat. Each t-SNE and UMAP plot is generated by randomly sampling 10,000 instructions from the associated dataset. We observe that the t-SNE and UMAP plot of MAGPIE encompasses the area covered by the plots of Alpaca, Evol Instruct, and UltraChat. This suggests that MAGPIE datasets provides a broader or more diverse range of topics, highlighting its extensive coverage across varied themes and subjects.

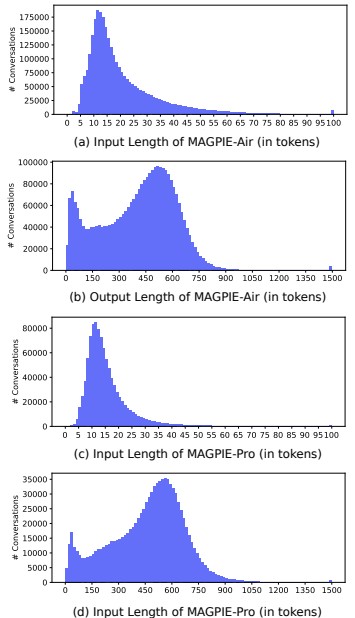

Figure 6: Lengths of instructions and responses in MAGPIE-Air/Pro.

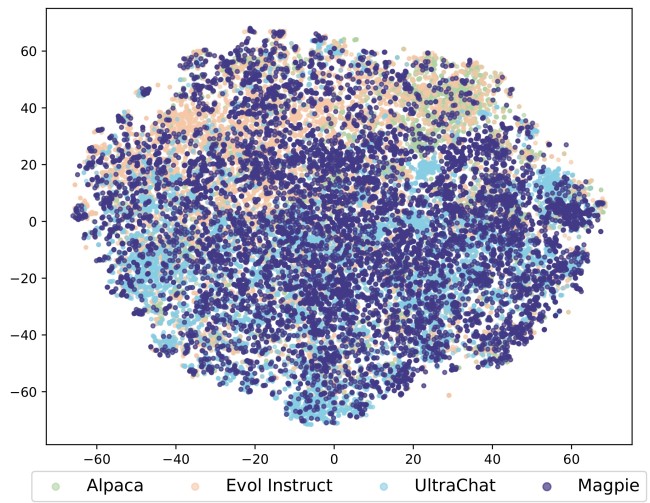

Figure 7: This figure compares the t-SNE plot of MAGPIE-Pro with those of Alpaca, Evol Instruct, and UltraChat, each of which is sampled with 10,000 instructions. The t-SNE plot of MAGPIE-Pro encompasses the area covered by the other plots, demonstrating the comprehensive coverage of MAGPIE-Pro.

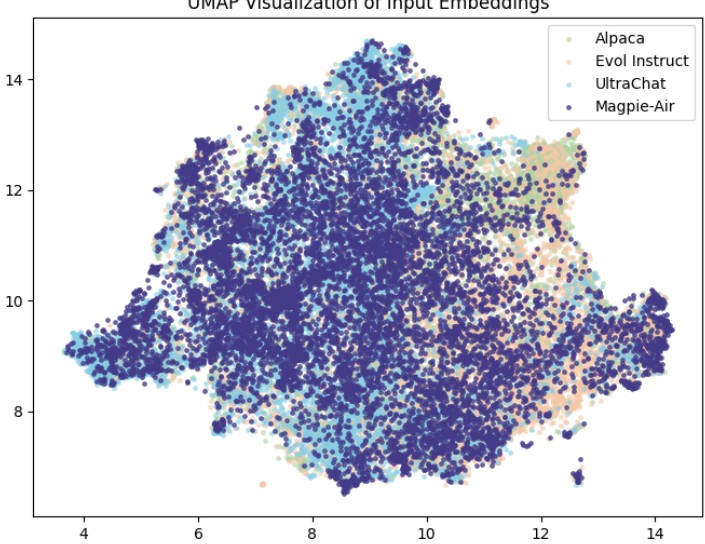

Figure 8: This figure compares the UMAP plot of MAGPIE-Air with those of Alpaca, Evol Instruct, and UltraChat, each of which is sampled with 10,000 instructions. The UMAP plot of MAGPIE-Air encompasses the area covered by the other plots, demonstrating the comprehensive coverage of MAGPIE-Air.

**Task Categories of MAGPIE-Pro and MAGPIE-Air.** Figure 9 illustrates the task category distributions for MAGPIE-Pro and MAGPIE-Air, as labeled by Llama-3-Instruct. We observe that the task category distributions of these two datasets are largely similar, however, MAGPIE-Pro exhibits a higher percentage of creative writing tasks.

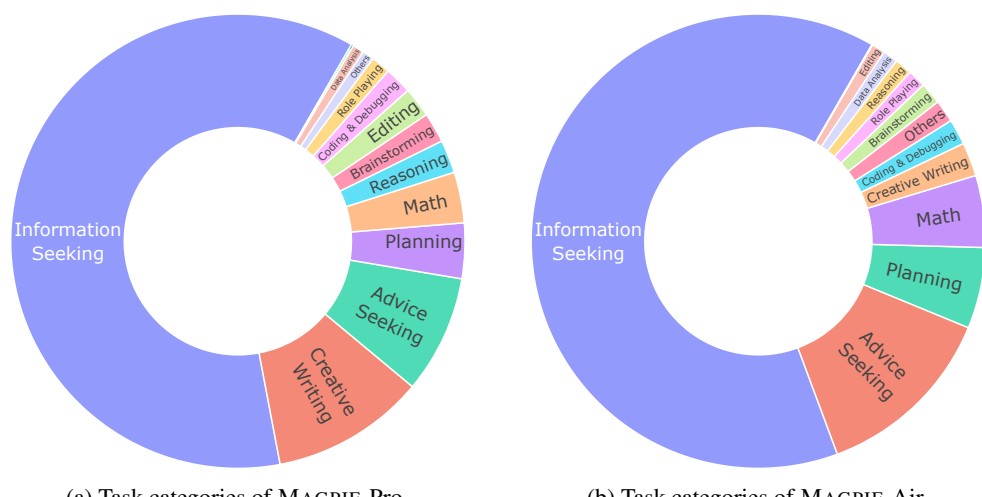

(a) Task categories of MAGPIE-Pro.          (b) Task categories of MAGPIE-Air.

Figure 9: This figure visualizes the task category of MAGPIE-Pro and MAGPIE-Air by topic tags.

**Topic Diversity of MAGPIE-Pro and MAGPIE-Air.** To validate the diversity of generated instructions, we conducted additional analysis using Topic Diversity metric from UltraChat (Ding et al., 2023). Our results are summarized in Table 6. The results demonstrate that our generated instructions are indeed more diverse in topic compared with other baselines.

Table 6: Comparison of Topic Diversity across Different Synthetic Datasets.

| Dataset | Alpaca | Evol Instruct | UltraChat | Magpie-Air | Magpie-Pro |
|---|---|---|---|---|---|
| Topic Diversity ($\downarrow$) | 0.13 | 0.09 | 0.10 | **0.05** | 0.06 |

**Visualization of Root Verbs and Their Direct Noun Objects.** Figure 10 visualizes the top common root verbs and their direct noun objects of MAGPIE-Air dataset. This indicates the diverse topic coverage of MAGPIE-Air.

### D.2 ADDITIONAL SAFETY ANALYSIS

Table 7 illustrates the percentage of different unsafe categories of MAGPIE-Air and MAGPIE-Pro, as labeled by Llama-Guard-2 (Team, 2024). We have two key observations. First, the proportion of data containing potentially harmful queries is minimal, with less than 1% for both datasets. Second, the majority of unsafe responses fall into the category of specialized advice, which includes responses that may offer specialized financial, medical, or legal advice, or suggest that dangerous activities or objects are safe.

Table 7: This table shows the percentage of different unsafe categories of MAGPIE-Air and MAGPIE-Pro tagged by Llama-Guard-2 Team (2024) model.

| Dataset | Safe | Violent Crimes | Non-Violent Crimes | Sex-Related Crimes | Child Sexual Exploitation | Specialized Advice | Privacy | Intellectual Property | Indiscriminate Weapons | Hate | Suicide & Self-Harm | Sexual Content | Others |
|---|---|---|---|---|---|---|---|---|---|---|---|---|---|
| MAGPIE-Air | 99.128% | 0.001% | 0.073% | 0.003% | 0.000% | 0.636% | 0.022% | 0.026% | 0.038% | 0.001% | 0.002% | 0.009% | 0.062% |
| MAGPIE-Pro | 99.347% | 0.001% | 0.049% | 0.002% | 0.000% | 0.446% | 0.015% | 0.074% | 0.014% | 0.001% | 0.004% | 0.011% | 0.036% |

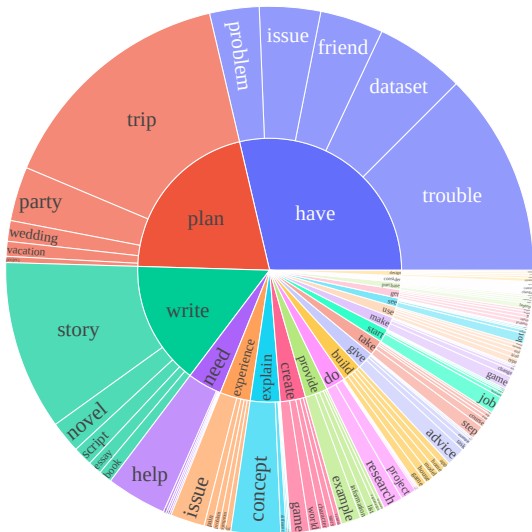

Figure 10: This figure demonstrates the top 20 most common root verbs (shown in the inner circle) and their top 5 direct noun objects (shown in the outer circle) within the MAGPIE-Air dataset. This indicates that MAGPIE encompasses a broad range of topics.

## D.3 ABLATION ANALYSIS ON GENERATION CONFIGURATIONS

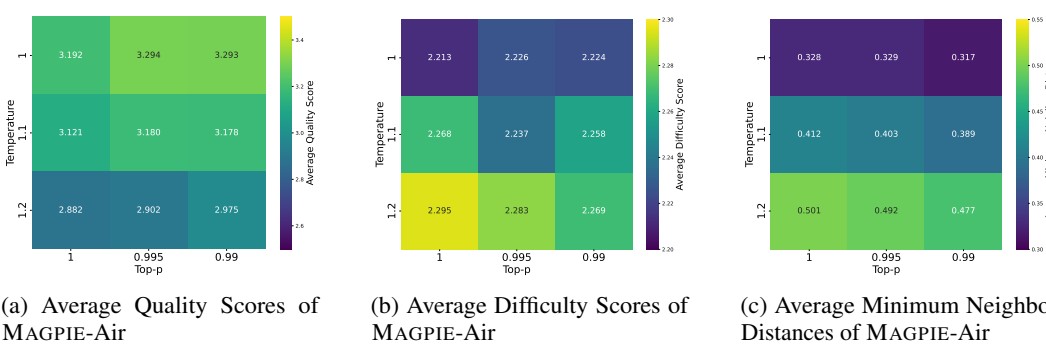

(a) Average Quality Scores of MAGPIE-Air

(b) Average Difficulty Scores of MAGPIE-Air

(c) Average Minimum Neighbor Distances of MAGPIE-Air

Figure 11: This figure illustrates the impact of varying decoding parameters on the quality, difficulty, and diversity of generated instructions. We observe that while higher temperature and top-p values may decrease the overall quality, they tend to increase both the difficulty and diversity of the instructions.

**Ablation Analysis on Decoding Parameters.** We conduct an ablation analysis on the decoding parameters used in generating instruction with MAGPIE. Specifically, we use three different temperatures (i.e., 1, 1.1, and 1.2) and top-p values (i.e., 1, 0.995, and 0.99) during Step 1 of MAGPIE. We use three metrics, **Average Quality Score**, **Average Difficulty Score** and **Average Minimum Neighbor Distance** to characterize the quality, difficulty, and diversity of instructions using different decoding parameters. The Average Quality Score is calculated by averaging the ratings of all data within a specific temperature-top-p pair, on a scale from 1 ('very poor') to 5 ('excellent'). Similarly, the Average Difficulty Score is rated on a scale from 1 ('very easy') to 5 ('very hard'). The Average Minimum Neighbor Distance is calculated by averaging the minimum neighbor distances, as defined in Section 3, for all data generated using the same decoding parameters.

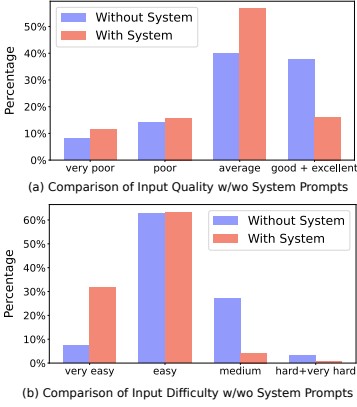

(a) Comparison of Input Quality w/wo System Prompts

(b) Comparison of Input Difficulty w/wo System Prompts

Figure 12: This figure compares the input quality and difficulty with and without system prompts.

The findings are summarized in Figure 11. We observe that higher temperature and top-p values may slightly decrease the overall quality of instructions, while simultaneously increasing the difficulty and remarkably enhancing the diversity of the instructions generated. The selection of these hyper-parameters should be tailored to the user's specific requirements, balancing the trade-offs between quality, difficulty, and diversity.

**Ablation Analysis on the System Prompt.** Figure 12 compares the use of system prompt compared with not using it in Step 1 of MAGPIE. Since the Llama-3 model does not have an official system prompt, we use the default system prompt from Vicuna (Chiang et al., 2023): A chat between a curious user and an artificial intelligence assistant. The assistant gives helpful, detailed, and polite answers to the user's questions. We observe that using a system prompt generally results in a decrease in the overall quality of instructions, and the instructions are easier. Consequently, we recommend not appending system prompts in default settings.

### D.4    IMPACT OF ANNOTATING MODELS

We note that LLMs may occasionally favor its own response (Deutsch et al., 2022). In what follows, we conduct experiments to evaluate the impact of annotating models when labeling quality and difficulty of the MAGPIE-Air dataset. We used the Qwen-2-7B-Instruct model (outside the Llama-3 family) to annotate the quality and difficulty of our MAGPIE-Air dataset. The statistics are summarized in Figure 13.

Our findings show that even when evaluated by Qwen-2-7B-Instruct, the MAGPIE-Air dataset maintains high quality and difficulty, which is even higher than those originally annotated by Llama-3-8B-Instruct. This suggests that our dataset's quality is robust across different annotators.

### D.5    CONTAMINATION ANALYSIS

(a) Statistics on Input Quality Using Different Annotators

(b) Statistics on Input Difficulty Using Different Annotators

Figure 13: This figure compares the impact of different annotators on evaluating the instruction quality and difficulty.

We conduct a contamination analysis of MAGPIE-generated instructions using the Alpaca Eval 2 and Arena Hard benchmarks. Following (Yu et al., 2024), we use an embedding model to convert both MAGPIE instructions and benchmark questions into embeddings, calculating cosine similarity scores to assess potential overlap.

Our analysis reveals that Arena Hard shows no evidence of data contamination, while some similarities were found between Alpaca Eval 2 and MAGPIE datasets generated by Llama-3, with a few entries showing cosine similarity greater than 0.9. Notably, the affected questions account for at most 10 out of 805 benchmark questions. This is expected, as Alpaca Eval data is constructed from a mixture of existing instruction datasets, many of which contain common questions such as "What's the capital of Australia?", "Can you code?", and "What's your name?" We note that even with this small degree of overlap, MAGPIE consistently outperformed the baselines, demonstrating its robustness and showing no significant impact on benchmark evaluations.

## E    DETAILED EXPERIMENTAL SETUPS

### E.1    EXPERIMENTAL SETUPS FOR GENERATING MAGPIE-AIR AND MAGPIE-PRO

As detailed in Appendix D.3, varying decoding parameters in Step 1 can significantly influence the quality, difficulty, and diversity of the generated instructions. To optimize the trade-offs among these attributes, we employ diverse decoding parameters for the generation of MAGPIE-Air and MAGPIE-Pro. Table 8 presents the configurations of MAGPIE-Air and MAGPIE-Pro, showcasing how diverse decoding parameters shape each dataset.

We employ greedy decoding to generate responses in Step 2 for MAGPIE-Air and MAGPIE-Pro. The intuition is that the word with the highest probability is more likely to originate from the model's training dataset.

Table 8: This table demonstrates the configurations of generating instructions of MAGPIE-Air and MAGPIE-Pro datasets with varying decoding parameters.

| Dataset | Decoding Parameters | | | Total #Convs |
|---|---|---|---|---|
| | Temperature | Top-p | #Convs | |
| MAGPIE-Air | 1.0 | 1.00 | 300K | |
| | 1.0 | 0.995 | 300K | |
| | 1.0 | 0.990 | 300K | |
| | 1.1 | 1.00 | 300K | |
| | 1.1 | 0.995 | 300K | |
| | 1.1 | 0.990 | 300K | |
| | 1.2 | 1.00 | 300K | 3M |
| | 1.2 | 0.995 | 300K | |
| | 1.2 | 0.990 | 300K | |
| | 1.25 | 1.00 | 100K | |
| | 1.25 | 0.995 | 100K | |
| | 1.25 | 0.990 | 100K | |
| MAGPIE-Pro | 1.0 | 1.00 | 300K | |
| | 1.1 | 0.995 | 300K | 1M |
| | 1.2 | 0.995 | 300K | |
| | 1.25 | 0.990 | 100K | |

## E.2 EXPERIMENTAL SETUPS FOR INSTRUCTION TUNING AND PREFERENCE TUNING

**Supervised Fine-Tuning Hyper-parameters.** Table 9 demonstrates the detailed supervised fine-tuning hyper-parameters. These experiments were conducted using Axolotl[5].

Table 9: This table shows the hyper-parameters for supervised fine-tuning.

| Hyper-parameter | Value |
|---|---|
| Learning Rate | $2 \times 10^{-5}$ |
| Number of Epochs | 2 |
| Number of Devices | 4 |
| Per-device Batch Size | 1 |
| Gradient Accumulation Steps | 8 |
| Effective Batch Size | 32 |
| Optimizer | `Adamw` with $\beta s = (0.9, 0.999)$ and $\epsilon = 10^{-8}$ |
| Learning Rate Scheduler | `cosine` |
| Warmup Steps | 100 |
| Max Sequence Length | 8192 |

**Preference Tuning Hyper-parameters.** Table 10 demonstrates the detailed DPO hyper-parameters for aligning Llama-3-8B using MAGPIE-Air-DPO and MAGPIE-Pro-DPO. These experiments were conducted using Alignment Handbook[6].

**Decoding parameters for evaluation benchmarks.** For Arena-Hard (Li et al., 2024b) and Wild-Bench (Lin et al., 2024), we follow its default setting and use greedy decoding for all settings. For AlpacaEval 2 (Li et al., 2023b) which allows the model provider to specify decoding parameters, we also employ greedy decoding in all experiments with a slightly increased repetition penalty ($RP = 1.2$) to mitigate the potential repetitive outputs during the generation.

---

[5]https://github.com/OpenAccess-AI-Collective/axolotl
[6]https://github.com/huggingface/alignment-handbook

Table 10: This table shows the hyper-parameters for direct preference optimization.

| Hyper-parameter | Value |
|---|---|
| Learning Rate | $5 \times 10^{-7}$ |
| Number of Epochs | 1 |
| Number of Devices | 4 |
| Per-device Batch Size | 2 |
| Gradient Accumulation Steps | 16 |
| Effective Batch Size | 128 |
| Optimizer | `Adamw` with $\beta s = (0.9, 0.999)$ and $\epsilon = 10^{-8}$ |
| Learning Rate Scheduler | `cosine` |
| Warmup Ratio | 10% |

# F  ADDITIONAL EXPERIMENTAL RESULTS

## F.1  PERFORMANCE OF MAGPIE-MT

Table 11 compares the performance of MAGPIE-Air-MT and MAGPIE-Pro-MT with their respective single-turn counterparts. We observe that the multi-turn datasets have enhanced performance, particularly in the Arena-Hard benchmark.

Table 11: This table compares the performance of the multi-turn versions, MAGPIE-Air-MT and MAGPIE-Pro-MT, with their single-turn counterparts. All models are instruction-tuned on the Llama-8B base models.

| Dataset | | AlpacaEval 2 | | | | | | Arena-Hard |
|---|---|---|---|---|---|---|---|---|
| | | GPT-4-Turbo (1106) | | | Llama-3-8B-Instruct | | | |
| | | LC (%) | WR (%) | SD | LC (%) | WR (%) | SD | WR (%) |
| MAGPIE-Air | Single-Turn | 22.66 | 23.99 | 1.24 | 49.27 | 50.80 | 1.44 | 14.9 |
| | MT | **22.98** | **24.02** | 1.27 | **49.63** | **51.42** | 1.40 | **15.5** |
| MAGPIE-Pro | Single-Turn | **25.15** | **26.50** | 1.30 | 50.52 | 52.98 | 1.43 | 18.9 |
| | MT | 24.21 | 25.19 | 1.28 | **52.92** | **54.80** | 1.41 | **20.4** |

## F.2  COMPARE MAGPIE AND SELF-INSTRUCT USING LLAMA-3-8B-INSTRUCT

To compare the performance of MAGPIE and other synthetic dataset generation methods using the same model, we follow the official Self-Instruct (Wang et al., 2023) setup and generate a 100K supervised fine-tuning dataset using Llama-3-8B-Instruct. For a fair comparison, we select the first 100K data samples from the MAGPIE-Air dataset generated by Llama-3-8B-Instruct. The performance of models fine-tuned with these two datasets is shown in the table 12.

Table 12: This table compares the performance of models fine-tuned using 100K instruction-following datasets generated by Self-Instruct and MAGPIE. All models are supervised-fine-tuned on the Llama-8B base models. We observe that MAGPIE significantly outperforms Self-Instruct across all benchmarks.

| Dataset | #Convs | AlpacaEval 2 | | | | | | Arena-Hard |
|---|---|---|---|---|---|---|---|---|
| | | GPT-4-Turbo (1106) | | | Llama-3-8B-Instruct | | | |
| | | LC (%) | WR (%) | SD | LC (%) | WR (%) | SD | WR (%) |
| MAGPIE-Air-100K | 100K | 20.17 | 21.33 | 1.21 | 46.82 | 48.76 | 1.44 | 15.7 |
| Self-Instruct (Llama-3) | 100K | 7.21 | 5.18 | 0.7 | 17.86 | 12.73 | 1.05 | 4.0 |

We observe a significant performance gap between models fine-tuned with datasets generated by Self-Instruct and our MAGPIE. Our analysis revealed that the instruction format in Self-Instruct-generated datasets is predominantly constrained by the patterns defined in the seed instructions, resulting in

a lack of diversity. This comparison indicates the novelty of our MAGPIE in generating diverse high-quality instructions without any seed questions.

## F.3  PERFORMANCE OF DOMAIN-SPECIFIC AND MULTILINGUAL MAGPIE DATASETS

**Domain Specific Data Evaluation.**    We choose code data as representative domain-specific data. We generate domain-specific data using the code instruction system prompt detailed in Appendix G. Using Qwen2.5-72B-Instruct as data generator, we create 100K synthetic code instructions via MAGPIE. We then fine-tune both Llama-3-8B base and Llama-3-8B-Instruct models using this dataset. The models are evaluated on HumanEval (Chen et al., 2021). The results shown in Table 13 demonstrate that our MAGPIE-generated code dataset effectively enhances Llama-3's performance on code-related tasks, validating MAGPIE's applicability to domain-specific instruction tuning.

Table 13: Performance Comparison on HumanEval.

| Alignment Setup | Pass@1 | Pass@10 | Pass@100 |
|---|---|---|---|
| Llama-3-8B-Instruct | 0.5574 | 0.7174 | 0.8049 |
| Llama-3-8B-base + MAGPIE-Code-100K | 0.5327 | 0.7134 | 0.8293 |
| Llama-3-8B-Instruct + MAGPIE-Code-100K | 0.5768 | 0.7334 | 0.8232 |

**Multilingual Data Evaluation.**    We evaluate MAGPIE's multilingual capabilities using Chinese as our representative language case. Following the method described in Section 2.2, we use Qwen2-72B-Instruct to generate 200K Chinese synthetic instructions. We then fine-tuned the Llama-3-8B base model with this dataset and evaluated its performance using multilingual MT-Bench. The results are presented in Table 14. The results demonstrate that models fine-tuned with our Chinese MAGPIE dataset outperform the official Llama-3-8B-Instruct on multilingual MT-Bench (zh-cn). This suggests MAGPIE's applicability to generate high-quality multilingual datasets.

Table 14: Performance Comparison on Chinese MT-Bench.

| Alignment Setup | Zh MT-Bench |
|---|---|
| Meta-Llama-3-8B-Instruct | 7.75 |
| Llama-3-8B-base + MAGPIE-Chinese-200K | 7.80 |
| Llama-3-8B-base + MAGPIE-Chinese-200K + MAGPIE-Pro-MT | 7.96 |

## F.4  ABLATION ANALYSIS ON DATA QUANTITY AND QUALITY

In what follows, we compare within the family of datasets generated by MAGPIE in Table 15. These datasets differ in size, deployment of filtering, and models used to generate data. We observe that as the dataset's size increases, the fine-tuned model's performance improves, indicating that data quantity plays a critical role in enhancing instruction-following capabilities. Furthermore, the model fine-tuned with MAGPIE-Pro-300K-Filtered outperforms those fine-tuned with the same or even higher amounts of raw data. This demonstrates the effectiveness of our filtering technique, and underscores the importance of data quality. Finally, we observe that the models fine-tuned with MAGPIE-Pro consistently outperform those fine-tuned with MAGPIE-Air. The reason is that MAGPIE-Pro is generated by the more capable model, i.e., Llama-3-70B-Instruct.

## F.5  ABLATION ANALYSIS ON FILTER DESIGNS

We conduct an ablation analysis on various filter designs within MAGPIE-Pro to assess their impact on the performance of supervised fine-tuned models. The results are presented in Table 16. We observe that different filtering strategies yield optimal performance on different benchmarks, and no single filter consistently achieves the best performance across all benchmarks. Therefore, determining how to select instructional data to enhance the performance in supervised fine-tuning is an interesting topic for future research.

Table 15: This table compares MAGPIE datasets within its family that differ in size, deployment of filtering, and models used to generate data. All models are supervised-fine-tuned on the Llama-8B base models.

| Dataset | | #Convs | AlpacaEval 2 | | | | | | Arena-Hard |
| | | | GPT-4-Turbo (1106) | | | Llama-3-8B-Instruct | | | |
| | | | LC (%) | WR (%) | SD | LC (%) | WR (%) | SD | WR(%) |
|---|---|---|---|---|---|---|---|---|---|
| MAGPIE-Air | 300K-Raw | 300K | 21.99 | 21.65 | 1.21 | 48.63 | 48.06 | 1.42 | 15.8 |
| | 3M-Raw | 3M | 22.96 | 21.09 | 1.20 | 50.57 | 48.40 | 1.42 | 16.1 |
| | 300K-Filtered | 300K | 22.66 | 23.99 | 1.24 | 49.27 | 50.8 | 1.44 | 14.9 |
| MAGPIE-Pro | 300K-Raw | 300K | 21.65 | 22.19 | 1.2 | 49.65 | 50.84 | 1.42 | 15.9 |
| | 1M-Raw | 1M | 24.16 | 23.93 | 1.25 | 49.97 | 50.34 | 1.43 | 16.7 |
| | 100K-Filtered | 100K | 20.47 | 24.52 | 1.25 | 47.92 | 52.75 | 1.43 | 17.2 |
| | 200K-Filtered | 200K | 22.11 | 26.02 | 1.26 | 51.17 | **56.76** | 1.41 | 15.9 |
| | 300K-Filtered | 300K | **25.08** | **29.47** | 1.35 | **52.12** | 53.43 | 1.44 | **18.9** |
| MAGPIE-Air + MAGPIE-Pro | 4M-Raw | 4M | 24.45 | 24.08 | 1.26 | 51.96 | 52.08 | 1.42 | 15.5 |

Table 16: This table compares the performance of different filter designs within MAGPIE-Pro. All models are supervised-fine-tuned on the Llama-8B base models.

| Dataset and Filter | | AlpacaEval 2 | | | | | | Arena-Hard |
| | | GPT-4-Turbo (1106) | | | Llama-3-8B-Instruct | | | |
| | | LC (%) | WR (%) | SD | LC (%) | WR (%) | SD | WR (%) |
|---|---|---|---|---|---|---|---|---|
| MAGPIE-Pro | Filter | 25.08 | **29.47** | 1.35 | 52.12 | 53.43 | 1.44 | **18.9** |
| | Filter 2 | **25.15** | 26.50 | 1.30 | 50.52 | 52.98 | 1.43 | **18.9** |
| | Filter 3 | 23.90 | 25.21 | 1.25 | 51.45 | 53.64 | 1.41 | 18.3 |
| | Filter 4 | 24.20 | 25.33 | 1.27 | **52.43** | 54.34 | 1.43 | 17.9 |
| | Filter 5 | 24.85 | 25.12 | 1.26 | 52.12 | 53.43 | 1.44 | 18.4 |
| | Filter 6 | 23.20 | 28.43 | 1.26 | 51.34 | **57.29** | 1.41 | 17.9 |

## F.6 ABLATION ANALYSIS ON RESPONSE GENERATOR

To investigate the impact of the response generator on the supervised fine-tuning performance using MAGPIE, we conduct an ablation study by replacing the response generator with Qwen-2-7B-Instruct (Yang et al., 2024a) within MAGPIE-Air-300K-Filtered. We note that the performance of Qwen-2-7B-Instruct is comparable to, or slightly weaker than, Llama-3-8B-Instruct. The results are summarized in Table 17.

We observe that although there is a slight performance degradation, the model fine-tuned using Qwen-2-7B-Instruct as the response generator still outperforms all baselines, including those using GPT-4 as the response generator. These findings indicate two key points: (1) The success of MAGPIE depends little on the specific response generator used, and (2) the instructions generated by MAGPIE are of high quality and diversity.

Table 17: This table compares the impact of different response generators on the model performance. All models are supervised-fine-tuned on the Llama-8B base models.

| Response Generator | AlpacaEval 2 | | | | | | Arena-Hard |
| | GPT-4-Turbo (1106) | | | Llama-3-8B-Instruct | | | |
| | LC (%) | WR (%) | SD | LC (%) | WR (%) | SD | WR (%) |
|---|---|---|---|---|---|---|---|
| Llama-3-8B-Instruct | 22.66 | 23.99 | 1.24 | 49.27 | 50.80 | 1.44 | 14.9 |
| Qwen2-7B-Instruct | 15.01 | 15.60 | 1.05 | 41.09 | 42.07 | 1.47 | 13.7 |

## F.7 TRUSTWORTHINESS OF MAGPIE-ALIGNED MODELS

In what follows, we conduct more experiments to compare MAGPIE model and Llama-3-8B-Instruct on the TrustLLM benchmark (Huang et al., 2024). The results for safety, fairness, ethics, privacy, and robustness are summarized in Table 18.

We observe that our supervised-fine-tuned model slightly underperforms Llama-3-8B-Instruct in terms of safety and fairness. However, it outperforms the official instruct model on ethics, privacy, and robustness. Considering that our fine-tuned model uses much fewer data samples (300K compared to over 10M), these results again highlight the high quality of data generated by MAGPIE.

Table 18: This table compares the performance of model supervised-fine-tuned using MAGPIE-Pro-300K-Filtered and the official Llama-3-8B-Instruct on the TrustLLM benchmark (Huang et al., 2024).

| TrustLLM | Evaluation/Dataset | Llama-3-8B-Instruct | MAGPIE-Pro-300K-Filtered |
|---|---|---|---|
| Safety | Jailbreak (RtA↑) | **0.93** | 0.80 |
| | Misuse (RtA↑) | **0.85** | 0.80 |
| | Exaggerated Safety (RtA↓) | **0.54** | 0.52 |
| Fairness | Stereotype Recognition (Acc↑) | **0.49** | 0.40 |
| | Stereotype Query Test (RtA↑) | **1.00** | 0.99 |
| | Disparagement Sex (p-value↑) | **0.99** | **0.99** |
| | Disparagement Race (p-value↑) | **0.55** | 0.47 |
| Ethics | Social Chemistry 101 (Acc↑) | **0.94** | 0.63 |
| | ETHICS (Acc↑) | 0.65 | **0.69** |
| | MoralChoice (Acc↑) | **0.97** | 0.95 |
| | MoralChoice (RtA↑) | 0.97 | **0.98** |
| Privacy | Privacy Awareness-Normal (RtA↑) | 0.33 | **0.71** |
| | Privacy Awareness-Augmented (RtA↑) | **1.00** | 0.98 |
| | Privacy Leakage (RtA↑) | 0.66 | **0.87** |
| Robustness | AdvGlue (RobustScore↑) | 0.42 | **0.58** |
| | OOD Detection (RtA↑) | **0.37** | 0.26 |
| | OOD Generalization (F1-Score↑) | 0.83 | **0.84** |

## F.8 IFEVAL EVALUATIONS OF MAGPIE-ALIGNED MODELS AND BASELINES

We compare the models fine-tuned with MAGPIE against baselines on IFEval Zhou et al. (2023b) using the LM-Evaluation-Harness framework Gao et al. (2024). The results are presented in Table 19.

Our results demonstrate that MAGPIE-generated datasets achieve comparable prompt-level and instruction-level strict accuracy scores to existing baseline datasets. Moreover, MAGPIE exhibits significantly higher performance in both prompt-level and instruction-level loose accuracy metrics. These findings indicate the high quality of MAGPIE-generated datasets.

Table 19: This table compares the performance of models fine-tuned using MAGPIE and other baseline datasets on the IFEval benchmark (Zhou et al., 2023b).

| Alignment Data | prompt_level_strict | inst_level_strict | prompt_level_loose | inst_level_loose |
|---|---|---|---|---|
| Self-Instruct (Llama-3) | 0.333 | 0.465 | 0.372 | 0.501 |
| ShareGPT | 0.331 | 0.454 | 0.372 | 0.492 |
| Evol Instruct | 0.344 | 0.463 | 0.377 | 0.494 |
| OpenHermes 1 | 0.340 | 0.453 | 0.377 | 0.488 |
| Tulu V2 Mix | 0.338 | 0.458 | 0.370 | 0.499 |
| WildChat | 0.372 | 0.489 | 0.423 | 0.538 |
| OpenHermes 2.5 | **0.381** | 0.493 | 0.436 | 0.536 |
| GenQA | 0.307 | 0.458 | 0.331 | 0.484 |
| Ultrachat | 0.298 | 0.421 | 0.346 | 0.466 |
| MAGPIE-Air-300K-Raw | 0.366 | 0.489 | 0.477 | 0.590 |
| MAGPIE-Air-300K-Filtered | 0.355 | 0.484 | 0.475 | 0.597 |
| MAGPIE-Air-300K-MT | 0.368 | **0.496** | **0.495** | **0.614** |
| MAGPIE-Pro-300K-Raw | 0.338 | 0.472 | 0.455 | 0.582 |
| MAGPIE-Pro-300K-Filtered | 0.298 | 0.432 | 0.401 | 0.529 |
| MAGPIE-Pro-300K-MT | 0.336 | 0.452 | 0.455 | 0.568 |

### F.9 ABLATION ANALYSIS ON THE IMPACT OF REWARD MODELS ON DPO PERFORMANCE

To investigate how the choice of reward model influences DPO performance, we performed an ablation study using two reward models: *RLHFlow/ArmoRM-Llama3-8B-v0.1* (Wang et al., 2024a) and *sfairXC/FsfairX-LLaMA3-RM-v0.1* (Xiong et al., 2024). Notably, ArmoRM exhibited stronger performance on RewardBench (Lambert et al., 2024). Following the procedure described in Section 2.2, we constructed DPO datasets based on scores from both reward models and evaluated their performance using AlpacaEval2 and Arena Hard benchmarks.

As shown in Table 20, our results demonstrate that employing a higher-performing reward model (ArmoRM) yields a better-performing LLM during DPO. This underscores the critical role of reward model quality in enhancing alignment and overall model performance.

Table 20: Comparison of Llama-3-8B DPO-trained models using different reward models. Metrics are AE2 LC, AE2 WR, and AH Score.

| Instruction Model | AE2 LC | AE2 WR | AH Score |
|---|---|---|---|
| Llama-3-8B-Air-DPO-ArmoRM | 45.48 | 50.43 | 35.9 |
| Llama-3-8B-Air-DPO-sfairXC | 37.87 | 45.08 | 33.1 |
| Llama-3-8B-Pro-DPO-ArmoRM | 50.10 | 53.53 | 35.7 |
| Llama-3-8B-Pro-DPO-sfairXC | 43.49 | 50.75 | 34.2 |

## G PROMPT TEMPLATES

### G.1 PROMPT TEMPLATES FOR MAGPIE EXTENSION

This section presents the prompt template used to generate MAGPIE-MT and control instruction tasks, as detailed in Figure 14 and Figure 15, respectively.

---

**Prompt for generating MAGPIE-MT**

<|begin_of_text|><|start_header_id|>system<|end_header_id|>

You are a helpful AI assistant. The user will engage in a multi–round conversation with you, asking initial questions and following up with additional related questions. Your goal is to provide thorough, relevant and insightful responses to help the user with their queries.<|eot_id|><|start_header_id|>user<|end_header_id|>

{instruction}<|eot_id|><|start_header_id|>assistant<|end_header_id|>

{response}<|eot_id|><|start_header_id|>user<|end_header_id|>

---

Figure 14: Prompt for generating MAGPIE-MT. We take Llama-3-8B-Instruct as an example. The placeholder `{instruction}` and `{response}` are from the first turn.

### G.2 PROMPT TEMPLATES FOR EVALUATION

Here, we present the prompt template employed to generate task categories, quality, and difficulty, as detailed in Figure 16, Figure 17, and Figure 18, respectively. The placeholder `input` represents the instructions to be evaluated.

## H EXAMPLE OUTPUTS OF MAGPIE MODELS

In the following Figure 19 and Figure 20, we present example outputs generated by the MAGPIE models and compare them with GPT-4-Turbo (1106), using Alpaca Eval 2 for evaluation.

**System Prompt Template**

<|begin_of_text|><|start_header_id|>system<|end_header_id|>

{System Prompt}<|eot_id|><|start_header_id|>user<|end_header_id|>

**System prompt for controlling math instruction tasks**

You are an AI assistant designed to provide helpful, step-by-step guidance on solving math problems. The user will ask you a wide range of complex mathematical questions. Your purpose is to assist users in understanding mathematical concepts, working through equations, and arriving at the correct solutions.

**System prompt for controlling code instruction tasks**

You are an AI assistant designed to provide helpful, step-by-step guidance on coding problems. The user will ask you a wide range of coding questions. Your purpose is to assist users in understanding coding concepts, working through code, and arriving at the correct solutions.

**System prompt for controlling translation tasks**

You are an AI assistant designed to provide accurate and contextually appropriate translations. Users will ask you to translate text between various languages. Your purpose is to assist users in understanding and conveying meaning across languages, maintaining the original context and nuances.

**System prompt for controlling multilingual instruction generation (Japanese + Math)**

あなたはAIアシスタントで、数学のを解くために役立つ、ステップバイステップのガイダンスを提供するようにされています。

Figure 15: Prompts for controlling instruction generation tasks. These examples illustrate how to guide Llama-3-8B-Instruct in generating instructions for specific domains: mathematics, coding, translation, and multilingual tasks. To adapt this approach for different instruction tasks, replace the System Prompt placeholder in the System Prompt Template with the appropriate domain-specific prompt.

Prompt for generating task categories

```
# Instruction
Please label the task tags for the user query.

## User Query
'''{input}'''

## Tagging the user input
Please label the task tags for the user query. You will need to analyze the user query and
     select the most relevant task tag from the list below.

all_task_tags = [
     "Information seeking", # Users ask for specific information or facts about various topics.
     "Reasoning", # Queries require logical thinking, problem–solving, or processing of
          complex ideas.
     "Planning", # Users need assistance in creating plans or strategies for activities and
          projects.
     "Editing", # Involves editing, rephrasing, proofreading, or other tasks related to the
          composition of general written content.
     "Coding & Debugging", # Users seek help with writing, reviewing, or fixing code in
          programming.
     "Math", # Queries related to mathematical concepts, problems, and calculations.
     "Role playing", # Users engage in scenarios requiring ChatGPT to adopt a character or
          persona.
     "Data analysis", # Requests involve interpreting data, statistics, or performing analytical
          tasks.
     "Creative writing", # Users seek assistance with crafting stories, poems, or other
          creative texts.
     "Advice seeking", # Users ask for recommendations or guidance on various personal or
          professional issues.
     "Brainstorming", # Involves generating ideas, creative thinking, or exploring possibilities.
     "Others" # Any queries that do not fit into the above categories or are of a miscellaneous
          nature.
]

## Output Format:
Note that you can only select a single primary tag. Other applicable tags can be added to
     the list of other tags.
Now, please output your tags below in a json format by filling in the placeholders in <...>:
'''
{{
     "primary_tag": "<primary tag>",
     "other_tags": ["<tag 1>", "<tag 2>", ... ]
}}
'''
```

Figure 16: Prompt for generating task categories

---

**Prompt for generating quality of instructions**

# Instruction
You need to rate the quality of the user query based on its clarity, specificity, and coherence.
The rating scale is as follows:

– very poor: The query is unclear, vague, or incoherent. It lacks essential information and
    context.
– poor: The query is somewhat unclear or lacks important details. It requires significant
    clarification.
– average: The query is moderately clear and specific. It may require some additional
    information for a complete understanding.
– good: The query is clear, specific, and mostly well–formed. It provides sufficient context for
    understanding the user's intent.
– excellent: The query is very clear, specific, and well–articulated. It contains all the
    necessary information and context for providing a comprehensive response.

## User Query
'''{input}'''

## Output Format
Given the user query, you first need to give an assessment, highlighting the strengths and/or
    weaknesses of the user query. Then, you need to output a rating from very poor to
    excellent by filling in the placeholders in [...]:
'''
{{
    "explanation": "[...]",
    "input_quality": "[very poor/poor/average/good/excellent]"
}}
'''
'''

Figure 17: Prompt for generating quality of instructions

---

**Prompt for generating difficulty of instructions**

# Instruction
You first need to identify the given user intent and then label the difficulty level of the user
    query based on the content of the user query.

## User Query
'''{input}'''

## Output Format
Given the user query, in your output, you first need to identify the user intent and the
    knowledge needed to solve the task in the user query. Then, rate the difficulty level of
    the user query as 'very easy', 'easy', 'medium', 'hard', or 'very hard'.

Now, please output the user intent and difficulty level below in a json format by filling in the
    placeholders in [...]:
'''
{{
    "intent": "The user wants to [....]",
    "knowledge": "To solve this problem, the models need to know [....]",
    "difficulty": "[very easy/easy/medium/hard/very hard]"
}}
'''

Figure 18: Prompt for generating difficulty of instructions

Question

Hi, I'm trying to solve a crossword puzzle, but I've never done one of these before. Can you help me out?

GPT-4-Turbo(1106)

Absolutely! I'd be delighted to help you with your crossword puzzle. Just tell me the clues and the number of letters needed for each answer (and any letters you may have already filled in), and I'll do my best to help you find the solutions. If you have any specific questions about how to approach solving crossword puzzles in general, feel free to ask those as well!

MAGPIE-Pro-DPO

# **Crossword Crusader's Guide**

Welcome to the world of crosswords! We're thrilled to assist you in navigating your first puzzle. This primer will cover the basics, tips, and tricks to get you started.

## **Understanding the Grid**

– A crossword puzzle consists of a grid with black squares (obstacles) and white squares (answer spaces).
– The grid is divided into rows and columns, usually labeled with numbers.
– Each answer spans one or more squares horizontally (across) or vertically (down).

## **Key Concepts**

1. **Clues**: These are the instructions to fill in the answers. Clues are usually written above (across) or to the left (down) of the answer space.
    – **Across clues** are read from left to right, like a sentence.
    – **Down clues** are read from top to bottom, like a column.
2. **Answer Length**: The number of squares an answer occupies is indicated by a digit(s) next to the clue. For example, "5–letter answer" or "Down: 7".
3. **Wordplay**: Crosswords often employ wordplay, which means clues might use:
    – **Anagrams**: Rearranged letters to form the answer.
    – **Homophones**: Words with similar sounds (e.g., "bank" and "bank").
    – **Double Definitions**: Two separate definitions for the same answer.
    – **Prefixes/Suffixes**: Clues might ask for a word with a specific prefix or suffix.

## **Tips for Solving**

1. **Start with easy clues**: Begin with shorter answers (3–5 letters) and work your way up.
2. **Fill in obvious answers**: Look for words that are easily recognizable or have only one possible answer.
3. **Use the grid**: Pay attention to the grid's structure. Look for patterns, symmetry, and blocked areas to help guide your answers.
4. **Check for intersections**: When filling in an answer, make sure it doesn't conflict with other answers at intersections.
5. **Don't be afraid to guess**: If you're unsure, try a few possibilities. You can always erase and try again.
6. **Use the "X" method**: When unsure, write an "X" in the square to indicate a possible answer. This helps you keep track of possibilities.
7. **Take breaks**: Crosswords can be mentally taxing. Take a break to refresh your mind.
8. **Consult the puzzle's instructions**: If you're stuck, check the puzzle's website or instructions for hints or special rules.

## **Additional Resources**

– If you're still unsure, try online crossword solvers like Crossword Solver, XWordInfo, or Cruciverb.
– Watch video tutorials or YouTube channels dedicated to crosswords, like Crossword Corner or The Crossword Guy.

## **Good Luck!**

Now, go ahead and dive into your crossword puzzle. Remember, the more you practice, the better you'll become. Have fun, and happy solving!

Figure 19: Comparison between outputs of MAGPIE-Pro-DPO and GPT-4-Turbo (1106) (Example 1).

Question

Do you know why turkeys became the official food of thanksgiving?

GPT-4-Turbo(1106)

The tradition of eating turkey for Thanksgiving is rooted in the history of the United States. There are a few reasons why turkey became the centerpiece of the Thanksgiving meal:

1. **Historical Accounts**:
   Although there is no definitive record that turkey was eaten at the very first Thanksgiving in 1621, when the Pilgrims celebrated a successful harvest with the Wampanoag Native Americans, it is known that wildfowl was part of the feast. Turkey, being a large bird native to North America, could feed a large group of people and was likely a convenient choice for such gatherings.

2. **Practicality**:
   Turkeys were abundant in North America and were a practical choice for a celebratory meal. Unlike chickens, which were more valuable for their eggs, and cows, which were more useful for their milk, turkeys were often kept for their meat and could be slaughtered without a significant loss to the homestead.

3. **Influence of Writings**:
   Sarah Josepha Hale, a prominent writer and editor in the 19th century, campaigned for the establishment of Thanksgiving as a national holiday. She published recipes and descriptions of Thanksgiving meals that included turkey, which may have influenced the tradition as Americans sought to replicate the ideal feast she depicted.

4. **Presidential Tradition**:
   In 1863, during the American Civil War, President Abraham Lincoln proclaimed Thanksgiving a national holiday to be celebrated on the last Thursday in November. As the holiday became more formalized, so did the menu, with turkey at the center.

5. **Symbolic and Practical Choice**:
   By the 19th and early 20th centuries, turkey had become the symbolic and practical choice for Thanksgiving. It was large enough to feed a crowd, relatively inexpensive, and not something that Americans would eat every day, which made it special for the occasion.

As a result of these factors, turkey became entrenched as the traditional main dish of Thanksgiving celebrations in the United States.

MAGPIE-Pro-DPO

A delightful question with a rich historical background. While not exclusively tied to a single event, the turkey's rise to Thanksgiving fame is a fascinating blend of cultural, economic, and symbolic factors. Here's the story:

## **Early American Tradition (1600s–1700s)**

In colonial New England, harvest festivals were common, celebrating the bounty of the land. These gatherings, often called "harvest home" or "thanksgiving," featured roasted meats, bread, and vegetables. Wildfowl, like ducks, geese, and even swans, were popular choices. Turkeys, although present, weren't the primary dish.

## **The Pilgrims' Influence (1621)**

The famous Pilgrim–Indian harvest feast at Plymouth Colony in 1621, often mislabeled as the "first Thanksgiving," played a significant role. While the menu is not fully documented, it's likely that wildfowl, including turkeys, were served, along with venison, corn, and other local foods. The Pilgrims, being English, were familiar with turkey consumption, which was already a staple in their homeland.

## **Economic and Ecological Factors (1700s–1800s)**

As the American colonies grew, so did the turkey population. The bird thrived in the New World's forests, where it was abundant and easy to hunt. This led to:

1. **Availability**: Turkeys were plentiful and relatively inexpensive, making them a practical choice for large gatherings.
2. **Dietary shift**: As the American diet transitioned from primarily fish and game to more domesticated animals, turkeys became a convenient, familiar, and nutritious alternative to beef and pork.
3. **Ecological advantage**: Turkeys were well–suited to the New World's climate and could be raised on small farms, reducing dependence on imported meats.

## **Symbolism and Cultural Evolution (1800s–1900s)**

The turkey's popularity at harvest festivals and family gatherings continued to grow. Over time, the bird became associated with:

1. **Abundance and Plenty**: Turkeys represented the harvest's bounty and the nation's prosperity.
2. **Family and Tradition**: The turkey became a symbol of togetherness and shared meals, which were essential to American culture.
3. **National Identity**: As the United States grew, the turkey's association with Thanksgiving helped create a sense of national identity and shared heritage.

## **The Official Food of Thanksgiving (1930s)**

In the 1930s, the turkey industry, which had been growing steadily, began to promote the bird as the quintessential Thanksgiving food. This effort was led by the National Turkey Federation (now the National Turkey Federation of America) and supported by the poultry industry, farmers, and food manufacturers.

In 1947, the National Turkey Federation officially declared the turkey the "traditional food of Thanksgiving." This declaration was not a government decree, but rather a cultural recognition of the bird's widespread acceptance and symbolic significance.

## **In Conclusion**

The turkey became the official food of Thanksgiving due to a combination of historical, economic, ecological, and cultural factors. Over time, the bird's availability, familiarity, and symbolism made it the perfect choice for a national harvest celebration. Today, the turkey remains an integral part of American Thanksgiving traditions. Happy Thanksgiving!

Figure 20: Comparison between outputs of MAGPIE-Pro-DPO and GPT-4-Turbo (1106) (Example 2).

# I   MAGPIE EXAMPLES

In this section, we demonstrate several representative instructions generated by MAGPIE with different task categories. The following instructions are generated by the Llama-3-70B-Instruct model.

---

**Information Seeking**

A few days ago, I was at a restaurant and I got a cup of coffee. However, when I went to take a sip, I realized it was a little too hot. So, I decided to let it cool down for a few minutes. As I waited, I noticed that coffee is actually two different colors. The part that was closest to the surface of the coffee is a lighter color, and the part that is deeper is a darker color. Have you ever observed this phenomenon before?

---

**Reasoning**

You have 3 switches but they are not labeled. Each switch corresponds to one of three light bulbs in a room. Each light bulb is either on or off. You can turn the switches on and off as many times as you want, but you can only enter the room one time to observe the bulbs. How can you figure out which switch corresponds to which light bulb?

---

**Planning**

You are the Founder of a Financial Planning Company. As a professional financial advisor, you are scheduled to meet a new client tomorrow. Specifically, what are you planning to do to prepare for this meeting?

---

**Editing**

What is the best way to re-write the sentence: "We call this the 'core' product or the 'core' offering" using proper quotation marks and avoiding the word "this"?

---

**Coding & Debugging**

Write a Python program that calculates the total cost of a customer's order. The program should ask for the customer's name, the number of items they want to purchase, and the price of each item. It should then calculate the total cost by multiplying the number of items by the price of each item and adding 8% sales tax. The program should display the customer's name, the number of items, the price of each item, and the total cost, including sales tax.

---

**Math**

In the following problem, please use integers to solve it. A water tank has 1000 L of water. On the first day, 1/5 of the water was drained. On the second day, 3/10 of the remaining water was drained. On the third day, 2/5 of the remaining water was drained. On the fourth day, 3/4 of the remaining water was drained. How many liters of water are left after the fourth day?

---

**Role Playing**

In this game, you will be the host, and I will be the contestant. You will ask me a series of questions, and I will try to answer them correctly. The questions will be multiple choice, and I will have a 25% chance of getting the correct answer if I just randomly guess. However, I am a clever contestant, and I will try to use logic and reasoning to increase my chances of getting the correct answer.

---

**Data Analysis**

The personnel manager at a company is tasked with finding the average salary of new hires. She has collected data on the salaries of 13 new hires. She wants to know if there is a statistical difference between the average salary of new hires and the national average salary. The national average salary is $45,000. The sample of new hires has a mean salary of $42,800 and a standard deviation of $4,200.

**Creative Writing**

Write a paragraph about a mythical creature that you created. The creature is small, no larger than a house cat. It has shimmering scales that reflect light, and can emit a soft, pulsing glow from its body. It has large, round eyes that seem to see right through you, but with a gentle kindness. It has a soft, melodious voice, and can communicate with humans through a form of telepathy.

**Advice Seeking**

How do you handle stress and overwhelm?

**Brainstorming**

Can you give me some ideas for a spontaneous, fun and memorable birthday celebration for my partner?

**Others**

What does "sdrawkcaB" mean?

MAGPIE can also generate domain-specific instructions using models that are tailored to particular fields, as mentioned in Section 2.2. The following instructions are generated by DeepSeek-Coder-V2 (Zhu et al., 2024) and Qwen2-Math-7B-Instruct (Yang et al., 2024a), respectively.

**DeepSeek-Coder-V2 (Code Instruction)**

You are given a list of emails. You need to write a Python function that returns the domain, excluding the @ symbol, for each email.

**Qwen2-Math-7B-Instruct (Math Instruction)**

A rectangle with length 12 units and width 8 units is scaled by a factor of 2 to form a new rectangle. Determine the dimensions of the new rectangle and calculate its area. Compare the area of the new rectangle to the area of the original rectangle.

We note that MAGPIE's capabilities extend beyond generating English datasets to producing diverse multilingual datasets. The following instructions are generated by the Qwen2-72B-Instruct model.

**Chinese**

从给定的两个整数中找到较大的一个。但是你不能使用任何比较操作符（如>, <, !=等）或数学运算符（如+, -, *, /等）来实现它。你只能使用位操作符和逻辑操作符。

**German**

Das Debate-System 'Oxford-Oberhaus' wird bei ersten Auseinandersetzungen verwendet. Bitte erklären sie, wie dieses System funktioniert.

**Spanish**

Según la encuesta anual de satisfacción al cliente que acabamos de realizar, parece que la satisfacción general de los clientes con nuestro rendimiento ha disminuido. ¿Podrías preparar una presentación detallada para la reunión del lunes que analice los resultados, identifique las áreas problemáticas y proporcione posibles soluciones basadas en los datos recogidos?

**Portuguese**

Ho comprato una nuova inchiostriera sulla quale è presente la scritta "Non manipolare". Cosa signidica?

**Italian**

Crie um exemplo de uma conversa entre dois personagens, um MC de hip hop e um pianista clássico, discutindo sobre seus estilos favoritos de música.

