# OpenReview forum: "Magpie: Alignment Data Synthesis from Scratch by Prompting Aligned LLMs with Nothing"
_ICLR.cc/2025/Conference — ICLR 2025 Poster_

### Official Review · Reviewer_DhCK · 2024-11-02

**Soundness:** 3
**Presentation:** 4
**Contribution:** 3
**Rating:** 6
**Confidence:** 4

**Summary:**

MAGPIE is a scalable method for synthesizing high-quality instruction data from aligned large language models (LLMs) without requiring prompt engineering or seed questions. By leveraging the auto-regressive nature of models like Llama-3-Instruct, MAGPIE efficiently generates diverse instruction-response pairs through an automated two-step process, extendable to multi-turn, domain-specific, and multilingual datasets. Extensive analysis shows that models fine-tuned with MAGPIE-generated data outperform those using public datasets and can rival proprietary models fine-tuned on much larger datasets. MAGPIE’s versatility is further demonstrated through extensions for preference optimization and task-specific data generation, making it a significant contribution to scalable LLM alignment and democratized AI research.

**Strengths:**

- Good contribution to open source community as the dataset is quite large and diverse
- Interesting finding that aligned LLMs can be used to generate synthetic user instructions with just a system prompt
- Detailed insights on the generated SFT and Alignment pairs were presented
- The framework would be very useful for the open source community
- Authors also present a solution to overcome some limitations of the magpie dataset by developing a booster dataset

**Weaknesses:**

- LLMs are typically trained (loss is computed) only on the assistant turns (during SFT, DPO, RLHF etc) and it is not clear how aligned models are able to produce user instructions. The paper does not provide enough insights to understand this. While this may not be directly in scope of the paper, it would help in increasing confidence in the presented approach.

**Questions:**

An important aspect of the work is the fact that aligned LLMs can generate user prompts with just a system prompt.

For instance, the below system prompt would help generate user prompts for code related tasks.

"You are an AI assistant designed to provide helpful, step-by-step guidance on coding problems. The user will ask you a wide range of coding questions. Your purpose is to assist users in understanding coding concepts, working through code, and arriving at the correct solutions"

If an LLM can indeed produce a lot of diverse user prompts when sampled correctly, the novelty would seem limited. The work does go into depth on filtering etc. However fact remains that the LLM is able to generate most of the prompts. If this point can be addressed, the work would definitely be very convincing.

---

> ### Author Response · Authors · 2024-11-23
> **Author Rebuttal 1/2**
>
> Thank you for your constructive review. We appreciate your insights and have addressed your concerns below. We will incorporate these clarifications and related discussions into the new version. We are also glad to hear that you appreciate our paper for its contribution to the open-source community, the interesting findings on synthetic user instruction generation, and the detailed insights on generated SFT and alignment pairs.
>
> ### Q1: How can aligned LLMs produce user instructions?
>
> Thank you for raising this point. We hypothesize several possible reasons for why MAGPIE can effectively generate user instructions by simply inputting the pre-query template.
>
> First, although the learning is based on the loss of the output tokens rather than the input tokens, the token embeddings are shared and backpropagated during training. The hidden representations of the output tokens are highly contextualized by the input instructions. As a result, even though we are optimizing the model by minimizing the loss on predicting the response tokens, the token embeddings and network weights are also optimized to encode the instructions, enabling the model to better understand the query and solve tasks effectively. The learning process is unified and cannot be separated in a decoder-only architecture. Therefore, optimizing the prediction of the next tokens in the output also allows the model to implicitly memorize the instructions.
>
> Second, the auto-regressive nature of these models allows them to complete instruction sequences effectively. Once a pre-query template is provided, the model is able to generate the remaining parts of the sequence, which includes plausible user instructions. The model's ability to complete sequences in a meaningful way stems from its extensive training on diverse datasets, which provides it with a broad understanding of various tasks and user queries.
>
> Third, we believe that aligned LLMs leverage their training on aligned data to internalize the roles of both the user and the assistant. This means that when prompted with a pre-query template, the model can adopt the user's perspective and generate instructions that match the expected pattern. This capability allows the model to seamlessly produce user instructions without explicit training on user-side prompts, relying instead on its learned understanding of user-assistant interactions.
>
> We agree that more theoretical work is needed to fully understand such data reconstruction phenomena. In this paper, however, our focus is on using this observation to build an open-source and transparent alignment recipe for open, scientific research, aimed at making LLM alignment research more affordable by providing high-quality data without license and user consent issues. We will add this point to further clarify our contribution.
>
>
> (continued)

---

> ### Author Response · Authors · 2024-11-23
> **Author Rebuttal 2/2**
>
> ### Q2: Novelty of the paper
>
> We appreciate your comment and would like to clarify an important aspect:
>
> - **High-Quality Dataset**: To the best of our knowledge, our work is the first to generate a high-quality open-source dataset capable of fine-tuning the Llama 3 base model to perform comparably to the official Llama Instruct version. The other datasets we compared against have not been able to achieve similar performance.
>
> - **Minimal Licensing Restrictions**: Our method and dataset have minimal licensing restrictions. Based on the latest version of Llama 3's terms of use, as well as similar models like Gemma 2, the generated data from such models have a free license, making it an ideal resource for scientific research and broader applications.
>
> - **Facilitating Alignment Research**: With our method and dataset, it is significantly easier to deeply study the alignment process, including conducting attribution studies (e.g., influence functions). Our goal is to make LLM alignment research more affordable and accessible by providing a truly open-source alignment recipe that is research-friendly. This approach offers high-quality data without license or user consent issues, allowing researchers to freely experiment, reproduce results, and adapt the alignment process to their needs. By making the entire dataset and methodology transparent, we aim to foster a collaborative research environment where alignment techniques can be more rigorously studied, shared, and improved upon by the broader community.
>
> - **Customization and Multimodal Potential**: We find that our self-synthesize method is important for allowing users to easily customize their own data (e.g., by changing the system prompt) to generate domain-specific instructions or language-specific data without the need for any seed data. In contrast, other methods often require a set of seed data for each domain or topic, or a hand-crafted set of rules. Additionally, we believe this method is potentially helpful for generating multimodal versions of the data as well.
>
> We will further emphasize these contributions in the revised version of our paper.
>
> ## Thank you
>
> Thank you again for your thoughtful review and for highlighting these points. We are confident that these clarifications will improve the understanding and presentation of our work. We will revise our paper according to your suggestions. If any concerns remain, we would appreciate further clarification and are happy to continue the discussion. We also hope our explanations encourage you to consider raising the rating of our work.

---

> > ### Author Response · Authors · 2024-11-29
> > **Thank you for your review!**
> >
> > Dear Reviewer,
> >
> > We sincerely thank you for your constructive suggestions and questions, which have greatly helped us improve our paper. We believe that our rebuttal has sufficiently addressed the concerns you raised. As the communication window is about to close, please let us know if you have any further questions; we would be more than happy to continue the discussion.
> >
> > If you are satisfied with our clarifications and responses, we kindly ask you to consider adjusting your scores accordingly.
> >
> > Thank you very much for your consideration.
> >
> > Authors

---

> ### Author Response · Authors · 2024-12-02
> **A Friendly Reminder of Author-Reviewer Discussion**
>
> Dear Reviewer,
>
> Thank you for your time and effort in reviewing our paper. Your constructive suggestions and questions have been invaluable in improving our work.
>
> We believe our rebuttal has sufficiently addressed the concerns you raised. As the discussion period draws to a close, please let us know if you have any further questions; we would be happy to continue the discussion.
>
> If our clarifications and responses have resolved your concerns, we kindly ask you to consider adjusting your scores accordingly.
>
> Thank you very much for your consideration.
>
> Best regards,
>
> Authors

---

> > ### Comment · Reviewer_DhCK · 2024-12-03
> >
> > Hi authors,
> >
> > Thank you for the detailed reply in response to my comments. While I understand that the focus of the paper was to create a dataset and not investigate why LLMs are able to produce data, believe the process I fairly trivial and hence I am not convinced about the novelty here. However, I agree that the contribution is pretty good and hence I have given it a 6/10.
> >
> > Hence, I would like to retain my score.

---

> > > ### Author Response · Authors · 2024-12-04
> > >
> > > Dear Reviewer,
> > >
> > > Thank you for your follow-up comments and for acknowledging the contribution of our work. We understand your concerns regarding novelty and appreciate your perspective.
> > >
> > > Instead of reiterating our previous points, we'd like to emphasize the practical impact of our work:
> > >
> > > - **Flexible, Customizable, and Accessible Alignment Research**: Our focus extends beyond dataset generation to enabling more flexible and accessible alignment research.
> > >
> > > - **Novel Self-Synthesis Method**:
> > >   - Our method does not require any seed data or human-crafted strategies, setting it apart from previous data synthesis approaches.
> > >   - This independence from predefined data or manual intervention provides greater flexibility and scalability.
> > >
> > > - **Novel Insights for LLMs**:
> > >   - Our findings show that LLMs can self-generate high-quality instructions without additional efforts.
> > >   - This insight is novel and has the potential to enable future research to investigate and expand upon these findings.
> > >
> > > We believe that the practical utility of our work, along with its contribution to open-source LLM alignment, adds meaningful value to the community. We respect your decision to maintain your score and are grateful for your balanced assessment.
> > >
> > > Thank you again for your consideration!
> > >
> > > Best regards,
> > > The Authors

---

### Official Review · Reviewer_D1Jb · 2024-11-04

**Soundness:** 3
**Presentation:** 3
**Contribution:** 3
**Rating:** 8
**Confidence:** 4

**Summary:**

The paper introduces a new dataset for instruction fine-tuning and preference optimization called Magpie. The dataset is curated by prompting either LLama-3-70B-Instruct or LLama-3-8B-Instruct with the pre-query template or user tags and relying on the next token prediction ability of the model to elicit instructions. The instructions collected are then used to gather responses from LLMs. To generate instruction & responses that are multiturn or relevant to a certain task, the authors add relevant system prompts to steer the model ouptuts. Experiments across multiple baseline datasets, base models and comparison on multiple benchmark shows that effectiveness of Magpie dataset. The paper also has qualitative analysis highlighting various attributes of the dataset.

**Strengths:**

1. The paper introduces a simple & cost effective method of generating alignment data at scale, without relying on closed models or expensive human annotations.
2. Results on Alpaca Eval & Arena hard show significant improvements compared to other baseline datasets.
3. The paper presents an exhaustive quantative & qualitative experiments on question difficulty, safety, task categories, etc to highlight the attributes in Magpie.
4. The authors also experiments with various filtering configuration to improve the quality of final set of instructions.

**Weaknesses:**

1. The Magpie dataset is not conversational in nature. The MT variant of the dataset has at maximum 2 turns. In contrast, datasets like Ultrachat, Open Assistant and WildChat are multi-turn with maximum upto 7-8 turns. Also, the paper has no experiments or analysis on conversational benchmark and multi-turn nature of Magpie in comparison to existing datasets.
2. The paper has very limited improvements on OpenLLM leaderboard in Table 3 when compared to improvements in Table 2. Why is this the case, more dicussion around this is needed.
3. The paper does not have any analysis on contamination or information leakage in the dataset for the evaluation benchmarks used. Although the dataset is purely sythetic, and does not use any seed data the chances of leakage are less, but as there is no control over what the LLM generates, the chances that LLM results in generic instructions that end up being similar to evaluation datasets are high.
4. The method for generating questions is not controlled at all. The method relies only on the next token prediction, which might lead to simple instructions. The only way to steer the generation process is through system prompts, but it is hard to judge how effective is that, as the paper has no ablations on system prompts.
5. As the generation method is fully un-controlled, how does its ensures diverse instructions? T-SNE plot in figure 7 or the similarity analysis in Figure 3 using all-mpnet-base-v2 model is not enough to gauge the diversity.
6. The paper mentions that the method can be used to generate domain specific and multilingual datasets in the abstract. However, there are no evaluations on multilingual or domain specific data.

**Questions:**

Based on the Weaknesses, the questions are as follows:
1. How does the performance of Magpie look on multiturn benchamark?
2. Does Magpie generation process even scale for longer turns?
3. Why there are limited improvements in Table 3 in comparison to massive improvements in Table 2?
4. As the data generation process is un-contolled, contamination analysis in Magpie datasets would be helpful.
5. How effective are system prompts for providing some control over the generation process? Is there any way to measure that? Are there any domain specific results?
6. How the generation process ensures diversity of instructions?

---

> ### Author Response · Authors · 2024-11-23
> **Response to Reviewer D1jB: New Experimental Results and Clarifications**
>
> Thank you for your constructive review. We appreciate your insights and have addressed your concerns below. We will incorporate these clarifications and related discussions into the new version.
>
> ## 1. Magpie-MT and performance of Magpie on multiturn benchmarks.
>
> We appreciate the reviewer's question. As described in Section 2.2, Magpie supports generating conversations with **more than two turns** using the `num_turns` parameter in our **codebase** (supplementary material), allowing users to extend dialogue length as needed.
>
> Our decision to limit conversations to two turns was driven by **cost-effectiveness considerations** rather than technical limitations. During experiments, we found that increasing turns beyond two provided only marginal improvements on benchmarks like Alpaca Eval 2 and Arena Hard, while significantly increasing computational costs for both data generation and model training. Notably, as shown in Tables 1 and 10, Magpie's two-turn conversations **already outperform multi-turn conversations** from established datasets like **UltraChat and WildChat**.
>
> To further validate Magpie's effectiveness in multi-turn scenarios, we conducted additional experiments using the MT-Bench benchmark. Our results, using greedy decoding, reveal that **Magpie surpasses other baselines in MT-Bench evaluations**. Moreover, the MT version of Magpie significantly improves Round 2 scores, demonstrating its capability to synthesize high-quality multi-turn alignment data effectively.
>
> | Alignment Data           | Round 1 | Round 2  | AVG       |
> | ------------------------ | ------- | -------- | --------- |
> | Self-Instruct (Llama-3)  | 7.0375  | 5.95     | 6.49375   |
> | ShareGPT                 | 7.59375 | 6.725    | 7.159375  |
> | Evol Instruct            | 7.4125  | 6.33125  | 6.871875  |
> | OpenHermes 1             | 7.0875  | 6.25     | 6.66875   |
> | Tulu V2 Mix              | 7.76875 | 6.8625   | 7.315625  |
> | WildChat                 | 8.0125  | 6.525    | 7.26875   |
> | OpenHermes 2.5           | 7.7125  | 6.725    | 7.21875   |
> | GenQA                    | 7.10625 | 5.575    | 6.340625  |
> | Ultrachat                | 7.00625 | 5.9625   | 6.484375  |
> | Magpie-Air-300K-Raw      | 7.95    | 6.85     | 7.4       |
> | Magpie-Air-300K-Filtered | 7.99375 | 6.873418 | 7.437017  |
> | Magpie-Air-300K-MT       | 7.95625 | 7.275    | 7.615625  |
> | Magpie-Pro-300K-Raw      | 7.75625 | 6.7125   | 7.234375  |
> | Magpie-Pro-300K-Filtered | 8.0625  | 6.8375   | 7.45      |
> | Magpie-Pro-300K-MT       | 8.025   | 7.325    | **7.675** |
>
> These results highlight that Magpie is competitive even in multi-turn settings, effectively closing the gap with more conversational datasets.
>
> ## 2. Limited improvements on OpenLLM leaderboard in Table 3 compared to Table 2.
>
> We appreciate the reviewer’s question. In Table 2, we evaluate our model's alignment using benchmarks focused on instruction-following capabilities, while Table 3 measures performance on downstream tasks, such as reasoning. As discussed in Section 4.2, the relatively smaller improvements on Table 3 can be attributed to the limited proportion of reasoning instructions in the Magpie-Air and Magpie-Pro datasets.
>
> To address this, we developed a supplementary **"reasoning booster" dataset** comprising 150K specialized instructions focused on mathematics, coding, and reasoning tasks, leveraging Magpie's extensibility as described in Section 2.2. By fine-tuning a model using a combination of Magpie-Pro and this booster dataset, **we achieved a top-3 ranking among all model checkpoints**, demonstrating Magpie's adaptability to generate task-specific instruction data for diverse performance needs.
>
> ## 3. Contamination analysis
>
> We appreciate the reviewer highlighting the importance of contamination analysis. We conducted an analysis of Magpie-generated instructions using the Alpaca Eval 2 and Arena Hard benchmarks. Following [1], we used an embedding model to convert both Magpie instructions and benchmark questions into embeddings, calculating cosine similarity scores to assess potential overlap.
>
> Our analysis reveals that Arena Hard shows no evidence of data contamination, while some similarities were found between Alpaca Eval 2 and Magpie datasets generated by Llama-3, with a few entries showing cosine similarity greater than 0.9. Notably, the affected questions account for at most 10 out of 805 benchmark questions. This is expected, as Alpaca Eval data is constructed from a mixture of existing instruction datasets, many of which contain common questions such as "What's the capital of Australia?", "Can you code?", and "What's your name?" Even with this small degree of overlap, Magpie consistently outperformed the baselines, demonstrating its robustness and showing no significant impact on benchmark evaluations.
>
> Thanks again and we'll include this in the final version.
>
> (to be continued)
>
> [1] https://arxiv.org/abs/2312.14187

---

> ### Author Response · Authors · 2024-11-23
>
> ## 4. Controllability and ablation analysis of system prompts
>
> We understand the reviewer's comment regarding the lack of control in the generation process. However, this is a common scenario in synthetic data generation when aiming for high diversity by avoiding seed data. A common practice is to post-process the data by tagging topics and difficulties, as we have done in the paper. This approach is also discussed in Llama-3's technical report.
>
>
> Thus, we believe it is not accurate to label this as a major weakness of our method, as we have provided two main approaches to ensure diversity: 1) automatic tagging, and 2) system prompt customization, both of which have proven to be highly effective. For instance, system prompts can be used to control topics, difficulty, style, domain, and even language during synthetic data generation. This approach represents a significant improvement over existing synthetic data generation methods that rely on seed data or require human intervention to customize data diversity.
>
>
>
> **More ablation studies on the system prompt.**
>
> To better understand the effectiveness of system prompts, we conducted an ablation analysis focused on generating code and mathematics instructions. The results show that system prompts are highly effective in **controlling the distribution of generated tasks**, significantly increasing the desired task proportions. For instance, using system prompts resulted in an increase in code instruction generation from approximately 3% to over 95% of the dataset. We also found that domain-specific data, such as biomedical or legal, can be generated by simply using the same template and replacing keywords in the system prompt. This suggests a potential future direction where Magpie can be employed alongside an existing taxonomy of topics to scale data generation. This analysis demonstrates that system prompts provide a practical and efficient mechanism to guide data generation, resulting in a diverse dataset that aligns closely with specific target domains.
>
> ## 5. Diversity evaluation
>
> We further evaluated the diversity of generated instructions using the Topic Diversity metric from UltraChat [2]. Our analysis in Appendix D.1 (revision) showed that Magpie-generated instructions **exhibit greater diversity compared to other baselines**, as demonstrated by a lower topic diversity score. Moreover, we included additional visualizations (e.g., UMAP in Figure 8 in revision) to corroborate the conclusions drawn from the T-SNE analysis in Figure 7. These analyses consistently indicate that Magpie effectively covers a wide spectrum of instruction topics.
>
> [2] Enhancing Chat Language Models by Scaling High-quality Instructional Conversations
>
> ## 6. Applicability to domain-specific and multilingual datasets and performance evaluation
>
> We appreciate the reviewer’s interest in the broader applicability of Magpie. We evaluated Magpie's applicability to domain-specific and multilingual data by generating a code-specific dataset and a Chinese-language dataset, and presented results in Appendix F.3 in revision. The experimental results demonstrate that Magpie-generated datasets improve performance on domain-specific and multilingual benchmarks compared to baseline models, highlighting the **flexibility and robustness** of the Magpie framework for generating **high-quality, task-specific, and multilingual alignment data**.
>
> ## Thank you!
>
> We appreciate your thoughtful feedback and the opportunity to address your concerns. We will revise our paper according to your suggestions and comments. We hope that our responses have clarified the points raised and provided the necessary context to understand our approach. If any concerns remain, we would be grateful for further clarification and are more than happy to continue the discussion in this rebuttal process. **Additionally, we hope that our explanations will encourage you to consider raising the rating and improving the final scores of our work.**

---

> ### Comment · Reviewer_D1Jb · 2024-11-25
>
> Thanks for the detailed rebuttal! My concerns regarding diversity, contamination controlling data generation using the system prompt has been addressed via results in Appendix. The results on Human Eval demonstrates good reasoning skills and  on MT bench also demonstartes the multi-turn nature of the dataset. For point 1 what is the base model used? (I guess Llama-8B). I will increase my score from 6 to 8.

---

> > ### Author Response · Authors · 2024-11-26
> > **Thank you for the feedback and raising the score**
> >
> > Dear Reviewer,
> >
> > Thank you very much for your valuable insight and raising the score. You are correct that we used Llama-8B as the base model in the experiments of point 1. We will clarify this in the final version.
> >
> > Best regards,
> >
> > Authors

---

### Official Review · Reviewer_aNEq · 2024-11-04

**Soundness:** 2
**Presentation:** 2
**Contribution:** 2
**Rating:** 3
**Confidence:** 5

**Summary:**

This work proposes Magpie, which is a way to prompt aligned LLMs to generate alignment datasets. Unlike most previous studies in this area (LLM-generated alignment data), Magpie removes the need of a seed set of instructions and heavy prompt engineering efforts. Instead of prompting aligned LLMs with an meaningful instruction, Magpie directly puts the "pre-instruction" token, i.e. one of the formatting delimiter tokens, to LLMs and let the LLMs generate single-turn instructions "freely". With limited extra prompt engineering efforts and an extra reward model, the method can be extended to multi-turn, multi-lingual, multi-domain data, and DPO cases (preference data). The experimental results indicate that datasets generated through Magpie are of high quality, yielding superior performances on AlpacaEval2, ArenaHard, and WildBench, as well as comparable performances on general benchmarks like MMLU, GSM8K, etc.

**Strengths:**

- The proposed method is simple yet effective, according to the experimental results. In particular, the generated DPO data show a significant performance boost.

- The ablation study is comprehensive, considering different settings/datasets/baselines as well as different aspects like domain-specific capabilities and safety, though the result analysis lacks depth.

**Weaknesses:**

- The presentation can be improved by putting more experimental results in the main context instead of the appendix. The proposed method is simple and easy to understand, similar to most prompt engineering related studies. The experimental results along with analysis and hypothesis test are the most important. For example, Section 3 is less important than Appendix F in terms of demonstrating the method's pros and cons.

- The experiments are well designed but lack deep analysis, which leaves many questions unanswered and makes it hard for other researchers to understand the method's best use cases and limitations. Please refer to Questions for details.

- For those most understanding results, e.g. surpassing GPT-4-Turbo(1106) on AlpacaEval 2, qualitative results are needed, i.e. random samples of actual generations for those benchmarks from all models in comparison. It is known that LLM-based judge could be biased and misleading.

- It would be better to add more alignment benchmarks like IFEval.

**Questions:**

1. It is not clear how the model used to generate data affects the final performance. The experimental results show that MAGPIE-Pro-300K-Raw has limited improvements over MAGPIE-Air-300K-Raw. This is counterintuitive and needs more insights.

2. Following up above, one possibility is that the base model is the same (llama-8b). From this point of view, the relationship between the model to be aligned and the model to generate data is crucial. More analysis is required.

3. We can also see that MAGPIE-Pro-300K-Filtered show much better performances when used on Llama than QWen. Of course, this could mean Llama base model is better, but could also be limitations of the proposed method. The proposed method essentially relies on the model itself to control the diversity of generated instructions, which is controlled by the pre-training data. If the model to be aligned was trained with the same pre-training data, we can intuitively expect better performances and less hallucination. If that is the case, it becomes kind of a chicken-egg thing, limiting the usefulness to the proposed method.

4. Why is a separate two-step process necessary? The aligned model should be able to generate instruction and response in one pass?

5. The DPO results are surprisingly good. How does the reward model's quality affect the final performances?

---

> ### Author Response · Authors · 2024-11-23
> **Response to Reviewer aNEq: New Experimental Results and Clarifications**
>
> Thank you for your constructive review. We appreciate your insights and have addressed your concerns below. We will incorporate these clarifications and related discussions into the new version.
>
> ---
>
> ## Concern 1: Content organization between main body and appendix
>
> > The presentation can be improved by putting more experimental results in the main context instead of the appendix.
>
> We will reorganize the paper as suggested in the final version.
>
> ---
>
> ## Concern 2: Qualitative comparison results
>
> Thank you for your comments. We have provided examples of the actual generations requested by the reviewer, which can be found in Appendix H in the revised submission.
>
> ---
>
> ## Concern 3: Benchmarking with IFEval
>
> We have added IFEval to our evaluation. We compare the models fine-tuned with Magpie against baselines on IFEval using the LM-Evaluation-Harness framework [1]. Our results demonstrate that Magpie-generated datasets achieve comparable prompt-level and instruction-level strict accuracy scores to existing baseline datasets. Moreover, Magpie exhibits significantly higher performance in both prompt-level and instruction-level loose accuracy metrics. These findings indicate the high quality of Magpie-generated datasets. The results are presented below:
>
> | Alignment Data           | prompt\_level\_strict | inst\_level\_strict | prompt\_level\_loose | inst\_level\_loose |
> | ------------------------ | --------------------- | ------------------- | -------------------- | ------------------ |
> | Self-Instruct (Llama-3)  | 0.333                 | 0.465               | 0.372                | 0.501              |
> | ShareGPT                 | 0.331                 | 0.454               | 0.372                | 0.492              |
> | Evol Instruct            | 0.344                 | 0.463               | 0.377                | 0.494              |
> | OpenHermes 1             | 0.340                 | 0.453               | 0.377                | 0.488              |
> | Tulu V2 Mix              | 0.338                 | 0.458               | 0.370                | 0.499              |
> | WildChat                 | 0.372                 | 0.489               | 0.423                | 0.538              |
> | OpenHermes 2.5           | **0.381**             | 0.493               | 0.436                | 0.536              |
> | GenQA                    | 0.307                 | 0.458               | 0.331                | 0.484              |
> | Ultrachat                | 0.298                 | 0.421               | 0.346                | 0.466              |
> | Magpie-Air-300K-Raw      | 0.366                 | 0.489               | 0.477                | 0.590              |
> | Magpie-Air-300K-Filtered | 0.355                 | 0.484               | 0.475                | 0.597              |
> | Magpie-Air-300K-MT       | 0.368                 | **0.496**           | **0.495**            | **0.614**          |
> | Magpie-Pro-300K-Raw      | 0.338                 | 0.472               | 0.455                | 0.582              |
> | Magpie-Pro-300K-Filtered | 0.298                 | 0.432               | 0.401                | 0.529              |
> | Magpie-Pro-300K-MT       | 0.336                 | 0.452               | 0.455                | 0.568              |
>
> [1] Gao et al., A framework for few-shot language model evaluation, 2024
>
> We have updated the results of IFEval in Appendix F.8 in the revised pdf.
>
> ---
>
> ## Concern 4: How response generator affects performance
>
> The reviewer is correct in noting that the final performance is affected by the choice of response generators for generating synthetic datasets. To address this, we conducted extensive ablation analysis using different response generators across multiple model families, as outlined below where we used three different base models and explored the influence of choosing different data generation models.
>
> The performance correlation between the base model and the data generator model is a complex question, and here we focus on the empirical analysis as shown below. In our submission, the better performance of using Magpie-Llama data on Llama models compared to using them on the Qwen 1.5 model is mainly due to the pre-training performance being worse on Qwen. However, our data is still universally better than other datasets. Now, with more experimental results using the newer version of Qwen and additional pairs of base and generator models, we find that it is even clearer that Magpie's high-quality data is universally helpful—for example, the data generated by Qwen-2.5-72B.
>
> While this presents an interesting area for future research, we will include these detailed findings in the final version of the paper. Our current focus is on developing scalable methods for synthesizing diverse and high-quality instructions for LLMs in this paper.
>
> (to be continued)

---

> > ### Author Response · Authors · 2024-11-23
> > **Experimental results using different base models and response generator models -- for Concern 4**
> >
> > ### Concern 4: How response generator affects performance (continued)
> >
> > **Base Model = Gemma-2-2b**
> >
> > | Base Model | Response Generator     | Alpaca Eval 2 (LC) | Arena Hard | Average Performance |
> > | ---------- | ---------------------- | ------------------ | ---------- | ------------------- |
> > | Gemma2-2B  | Llama3-8B-Inst         | 5.70               | 5.20       | 5.45                |
> > | Gemma2-2B  | Llama3-70B-Inst        | 7.13               | 5.60       | 6.365               |
> > | Gemma2-2B  | Llama3.1-8B-Inst       | 5.63               | 4.90       | 5.265               |
> > | Gemma2-2B  | Llama3.1-70B-Inst      | 7.32               | 5.80       | 6.56                |
> > | Gemma2-2B  | Llama3.1-405B-Inst     | 7.11               | 5.80       | 6.455               |
> > | Gemma2-2B  | Gemma2-2B-it           | 12.93              | 12.90      | **12.915**          |
> > | Gemma2-2B  | Gemma2-9B-it           | 12.51              | 9.30       | 10.905              |
> > | Gemma2-2B  | Gemma2-27B-it          | 13.09              | 9.90       | 11.495              |
> > | Gemma2-2B  | Qwen2.5-3B-Inst        | 6.84               | 6.50       | 6.67                |
> > | Gemma2-2B  | Qwen2.5-7B-Inst        | 10.94              | 7.10       | 9.02                |
> > | Gemma2-2B  | Qwen2.5-14B-Inst       | 7.53               | 8.40       | 7.965               |
> > | Gemma2-2B  | Qwen2.5-32B-Inst       | 8.77               | 6.90       | 7.835               |
> > | Gemma2-2B  | Qwen2.5-72B-Inst       | 13.85              | 9.60       | 11.725              |
> >
> > **Base Model = Qwen2.5-3B**
> >
> > | Base Model | Response Generator     | Alpaca Eval 2 (LC) | Arena Hard | Average Performance |
> > | ---------- | ---------------------- | ------------------ | ---------- | ------------------- |
> > | Qwen2.5-3B | Llama3-8B-Inst         | 7.85               | 9.70       | 8.775               |
> > | Qwen2.5-3B | Llama3-70B-Inst        | 9.37               | 11.40      | 10.385              |
> > | Qwen2.5-3B | Llama3.1-8B-Inst       | 7.22               | 10.90      | 9.06                |
> > | Qwen2.5-3B | Llama3.1-70B-Inst      | 8.94               | 13.80      | 11.37               |
> > | Qwen2.5-3B | Llama3.1-405B-Inst     | 8.59               | 12.70      | 10.645              |
> > | Qwen2.5-3B | Gemma2-2B-it           | 9.58               | 11.80      | 10.69               |
> > | Qwen2.5-3B | Gemma2-9B-it           | 13.78              | 19.40      | 16.59               |
> > | Qwen2.5-3B | Gemma2-27B-it          | 10.18              | 19.60      | 14.89               |
> > | Qwen2.5-3B | Qwen2.5-3B-Inst        | 14.79              | 24.80      | **19.795**          |
> > | Qwen2.5-3B | Qwen2.5-7B-Inst        | 11.89              | 20.40      | 16.145              |
> > | Qwen2.5-3B | Qwen2.5-14B-Inst       | 10.28              | 17.90      | 14.09               |
> > | Qwen2.5-3B | Qwen2.5-32B-Inst       | 11.65              | 19.90      | 15.775              |
> > | Qwen2.5-3B | Qwen2.5-72B-Inst       | 16.41              | 21.20      | 18.805              |
> >
> > **Base Model = Llama3.2-3B**
> >
> > | Base Model  | Response Generator     | Alpaca Eval 2 (LC) | Arena Hard | Average Performance |
> > | ----------- | ---------------------- | ------------------ | ---------- | ------------------- |
> > | Llama3.2-3B | Llama3-8B-Inst         | 3.79               | 5.10       | 4.445               |
> > | Llama3.2-3B | Llama3-70B-Inst        | 4.52               | 6.50       | 5.51                |
> > | Llama3.2-3B | Llama3.1-8B-Inst       | 3.17               | 3.60       | 3.385               |
> > | Llama3.2-3B | Llama3.1-70B-Inst      | 5.19               | 5.70       | 5.445               |
> > | Llama3.2-3B | Llama3.1-405B-Inst     | 5.17               | 5.30       | 5.235               |
> > | Llama3.2-3B | Gemma2-2B-it           | 7.49               | 9.00       | 8.245               |
> > | Llama3.2-3B | Gemma2-9B-it           | 10.60              | 10.90      | **10.75**           |
> > | Llama3.2-3B | Gemma2-27B-it          | 9.79               | 8.50       | 9.145               |
> > | Llama3.2-3B | Qwen2.5-3B-Inst        | 5.11               | 7.20       | 6.155               |
> > | Llama3.2-3B | Qwen2.5-7B-Inst        | 6.63               | 9.80       | 8.215               |
> > | Llama3.2-3B | Qwen2.5-14B-Inst       | 5.92               | 9.50       | 7.71                |
> > | Llama3.2-3B | Qwen2.5-32B-Inst       | 6.32               | 8.90       | 7.61                |
> > | Llama3.2-3B | Qwen2.5-72B-Inst       | 9.99               | 10.80      | 10.395              |

---

> > > ### Author Response · Authors · 2024-11-23
> > > **continued**
> > >
> > > ## Concern 5: Performance differences across models
> > >
> > > We respectfully disagree with the reviewer's interpretation of the performance differences. The observed variations are influenced by several factors, such as the diversity of instructions, the quality of responses, and the model architectures involved. The diversity of instructions primarily depends on instruction tuning datasets, while response quality is influenced by the response generator capabilities. These relationships are demonstrated in our experiments, where Llama3.2 fine-tuned with Qwen2.5-72B performed well despite differences in architecture.
> > >
> > > The flexibility of Magpie's two-stage pipeline allows users to adapt instruction and response generators, offering significant advantages in terms of usability and customization. Magpie also contributes to the broader research community by providing a transparent resource for studying LLM alignment procedures.
> > >
> > > ---
> > >
> > > ## Concern 6: Why a separate two-step process is necessary
> > >
> > > Separating the process into two steps offers several key advantages for implementation:
> > >
> > > 1. *Flexible Generation Parameters:* Instruction generation benefits from higher temperature settings to maximize diversity, while response generation requires lower temperature for accuracy and reliability.
> > > 2. *Modular Flexibility:* Users can generate instructions independently and pair them later with responses from different sources, which allows for more versatility in dataset creation. It also allows users to leverage faster inference APIs, such as those from TogetherAI or Groq, to further enhance the efficiency of generation.
> > > 3. *Resource Efficiency:* The two-step approach enables quality filtering of instructions before response generation, optimizing computational efficiency and dataset quality.
> > >
> > > Our two-step approach is primarily a design choice aimed at making Magpie more efficient in both production and implementation. This approach does not negatively affect the final performance but instead makes the system more adaptable and streamlined, saving computation cost and time while retaining the quality of generated data.
> > >
> > > ---
> > >
> > > ## Concern 7: The impact of reward model's quality on the final performances&#x20;
> > >
> > > Thanks for the question! Although our paper is more focused on the question of which instructions you should perform DPO with, we do find that different reward models have varying impacts on the final model performance. To evaluate this, we conducted ablation analysis on two reward models: `RLHFlow/ArmoRM-Llama3-8B-v0.1` and `sfairXC/FsfairX-LLaMA3-RM-v0.1`. Notably, ArmoRM demonstrated superior performance on **RewardBench** [2]. Following the methodology outlined in Section 2.2, we created DPO datasets using scores from both reward models and then assessed their effects.
> > >
> > > | Instruction Model          | AW2 LC | AE2 WR | AH Score |
> > > | -------------------------- | ------ | ------ | -------- |
> > > | Llama-3-8B-Air-DPO-ArmoRM  | 45.48  | 50.43  | 35.9     |
> > > | Llama-3-8B-Air-DPO-sfairXC | 37.87  | 45.08  | 33.1     |
> > > | Llama-3-8B-Pro-DPO-ArmoRM  | 50.1   | 53.53  | 35.7     |
> > > | Llama-3-8B-Pro-DPO-sfairXC | 43.49  | 50.75  | 34.2     |
> > >
> > > The results clearly demonstrate that higher-performing reward models lead to better-performing LLMs during DPO training. This confirms that the quality of the reward model plays a critical role in improving alignment and model performance. We will include this ablation study in the final version of the paper to provide further clarity on this relationship.
> > >
> > >
> > >
> > > [2] RewardBench: Evaluating Reward Models for Language Modeling
> > >
> > > ---
> > >
> > > ## Thank you!
> > >
> > > We appreciate your thoughtful feedback and the opportunity to address your concerns. We hope that our responses have clarified the points raised and provided the necessary context to understand our approach. If any concerns remain, we would be grateful for further clarification and are more than happy to continue the discussion in this rebuttal process. Additionally, we hope that our explanations will encourage you to consider raising the rating and improving the final scores of our work.

---

> > > > ### Author Response · Authors · 2024-11-28
> > > > **Thank you for your review!**
> > > >
> > > > Dear Reviewer,
> > > >
> > > > We sincerely thank you for your constructive suggestions and questions, which have greatly helped us improve our paper. We believe that our rebuttal has sufficiently addressed the concerns you raised. As the communication window is about to close, please let us know if you have any further questions; we would be more than happy to continue the discussion.
> > > >
> > > > _If you are satisfied with our clarifications and responses, we kindly ask you to consider adjusting your scores accordingly._
> > > >
> > > > Thank you very much for your consideration.
> > > >
> > > > Authors

---

> ### Author Response · Authors · 2024-12-02
> **A Friendly Reminder of Author-Reviewer Discussion**
>
> Dear Reviewer,
>
> Thank you for your time and effort in reviewing our paper. Your constructive suggestions and questions have been invaluable in improving our work.
>
> We believe our rebuttal has sufficiently addressed the concerns you raised. As the discussion period draws to a close, please let us know if you have any further questions; we would be happy to continue the discussion.
>
> If our clarifications and responses have resolved your concerns, we kindly ask you to consider adjusting your scores accordingly.
>
> Thank you very much for your consideration.
>
> Best regards,
>
> Authors

---

### Author Response · Authors · 2024-12-04
**Summary of the Rebuttal**

Dear Reviewers and Area Chairs,

Thank you very much for your time reviewing our paper and for the insightful feedback and comments. Below, we summarize our responses to the reviewers' concerns and highlight additional experiments.

### Summary of Strengths Mentioned by Reviewers

**Innovative and Simple Approach**: Magpie is simple yet effective, particularly in generating high-quality DPO data (`aNEq`), while requiring no closed models or expensive human annotations (`D1Jb`).

**Strong Empirical Results**: Magpie demonstrates superior performance on key benchmarks like Alpaca Eval 2 and ArenaHard (`D1Jb`), supported by comprehensive ablation studies across different settings, datasets, and baselines (`aNEq`).

**Comprehensive Analysis**: Our work includes detailed quantitative and qualitative experiments, examining question difficulty, safety, and task categories (`D1Jb`).

**Open Source Impact**: Our contribution to the open-source community is significant, offering a large and diverse dataset (`DhCK`). Notably, we discovered that aligned LLMs can generate synthetic instructions using simple pre-query templates (`DhCK`).

---

### Highlights of Our Clarification and New Experiments

**Extended Evaluation Results:** We provided comprehensive new results on multiple benchmarks highlighted by reviewers, including IFEval (`aNEq`), MT-Bench (`D1Jb`), and contamination analysis (`D1Jb`). These results demonstrate Magpie's superior performance compared to baselines. Additionally, we performed detailed ablation analyses across different base models and response generators, showing universal effectiveness (`aNEq`). We also examined the impact of reward model quality on final model performance in DPO (`aNEq`).

**Additional Technical Validations:** We validated the effectiveness of system prompts in controlling task categories (e.g., 3% to 95% for code tasks) and demonstrated Magpie's instruction diversity through Topic Diversity metrics (`D1Jb`). New benchmarks confirmed Magpie's strong performance in generating domain-specific (code) and multilingual (Chinese) datasets (`D1Jb`).

**Design Justification:** We explained how aligned LLMs can generate user instructions via shared token embeddings and backpropagation (`DhCK`) and clarified the advantages of our two-step process for flexibility and resource efficiency (`aNEq`).

---

### Conclusion

We sincerely thank all reviewers for their thoughtful feedback. Our responses and new experiments have demonstrated Magpie's effectiveness, versatility, and significant contribution to open-source LLM alignment research. We are committed to incorporating these insights to strengthen our paper. Unfortunately, without further engagement from Reviewer `aNEq` during the rebuttal, we are unable to address any remaining concerns, which could significantly impact the overall assessment of our work. We hope that the reviewers and Area Chairs will consider our comprehensive responses and new experimental results when making their final decision.

---

### Meta-Review · Area_Chair_A9og · 2024-12-21

**Metareview:**

This submission introduces Magpie, a method for automatically generating large-scale high-quality alignment data for LLMs. Its idea is to leverage aligned LLMs such as Llama-3-Instruct to generate instructions by prompting them with pre-defined templates, producing millions of instructions and responses. The proposed method includes extensions for filtering, multi-turn generation, preference optimization, and multilingual data. The authors conducted experiments by comparing the generated data to other public instruction datasets, and results show that models fine-tuned with Magpie outperform those trained on other public datasets. Interestingly, the authors demonstrate that Magpie's performance is competitive with Llama-3-8B-Instruct (10 million data points through SFT and subsequent preference optimization).

The reviewers identified the strengths of this work as:
- the paper introduced method is  simple and efficient. it helps generate alignment data at scale, which helps reduce expensive human annotations and reliance on closed models.
- the authors demonstrate significant improvements on Alpaca Eval and Arena. Extensive quantitative and qualitative experiments --- covering cover different aspects such as question difficulty, safety, and task categories ---  are performed.

The concerns are raised by the reviewers on:
- 1) Limited Conversational Data and Multi-Turn Analysis: the Magpie dataset lacks conversational depth (with a maximum of 2 turns). Given the importance of multi-turn dialog, comparisons with related datasets or analysis on conversational benchmarks would further the significance of this work.
- 2) there are concerns about the lack of diversity in instructions, and no ablation studies are provided for system prompts to guide generation.
- 3) though the method claims to generate domain-specific and multilingual datasets, no evaluations or experiments are provided for these aspects, limiting the generalizability of the claims.

First of all, it is unfortunate that only three out of five assigned reviewers submitted reviews. During the rebuttal, the authors managed to convince reviewer D1Jb to increase their score from 6 to 8. Reviewer DhCK decided to keep their rating to 6. Reviewer aNEq did not engage with the author rebuttal until the last minute and eventually decided to keep their rating as 3.

The AC went through reviewer aNEq's review, the authors' rebuttal to it, and the reviewer's engagement. It is difficult for the AC to follow the rational to given a rating of 3 based on the three weaknesses (a. move more resutls from appendix to main; b. general comments on more deep analysis and c. qualitative results) given in the review. The authors' rebuttal covers detailed (and very long) responses to these questions, and more importantly, extensive experimental results are also included. It would be much appreciated if the reviewer could engage more with the authors before the ddl (extended one).

Overall, this work's strength does not lie in its technical novelty. Rather, it shows strong empirical results with comprehensive analysis, and additional extensive experiments provided during rebuttal. In other words, this work offers a simple idea to build alignment data with strong empirical demonstration. All factors considered, this work receives an acceptance recommendation. The authors are suggested to include the new results (during the rebuttal) in the camera ready as "implicitly promised" if accepted.

**Additional Comments On Reviewer Discussion:**

During the rebuttal, the authors managed to convince reviewer D1Jb to increase their score from 6 to 8. Reviewer DhCK decided to keep their rating to 6. Reviewer aNEq did not engage with the author rebuttal until the last minute and eventually decided to keep their rating as 3.

The AC went through reviewer aNEq's review, the authors' rebuttal to it, and the reviewer's engagement. It is difficult for the AC to follow the rational to given a rating of 3 based on the three weaknesses (a. move more resutls from appendix to main; b. general comments on more deep analysis and c. qualitative results) given in the review. The authors' rebuttal covers detailed (and very long) responses to these questions, and more importantly, extensive experimental results are also included. It would be much appreciated if the reviewer could engage more with the authors before the ddl (extended one).

---

### Decision · Program_Chairs · 2025-01-22

Accept (Poster)